# Faster Relative Entropy Coding with Greedy Rejection Coding

**Gergely Flamich**[*]
Department of Engineering
University of Cambridge
gf332@cam.ac.uk

**Stratis Markou**[*]
Department of Engineering
University of Cambridge
em626@cam.ac.uk

**José Miguel Hernández Lobato**
Department of Engineering
University of Cambridge
jmh233@cam.ac.uk

## Abstract

Relative entropy coding (REC) algorithms encode a sample from a target distribution $Q$ using a proposal distribution $P$ using as few bits as possible. Unlike entropy coding, REC does not assume discrete distributions or require quantisation. As such, it can be naturally integrated into communication pipelines such as learnt compression and differentially private federated learning. Unfortunately, despite their practical benefits, REC algorithms have not seen widespread application, due to their prohibitively slow runtimes or restrictive assumptions. In this paper, we make progress towards addressing these issues. We introduce Greedy Rejection Coding (GRC), which generalises the rejection based-algorithm of Harsha et al. (2007) to arbitrary probability spaces and partitioning schemes. We first show that GRC terminates almost surely and returns unbiased samples from $Q$, after which we focus on two of its variants: GRCS and GRCD. We show that for continuous $Q$ and $P$ over $\mathbb{R}$ with unimodal density ratio $dQ/dP$, the expected runtime of GRCS is upper bounded by $\beta D_{\mathrm{KL}}[Q\|P] + \mathcal{O}(1)$ where $\beta \approx 4.82$, and its expected codelength is optimal. This makes GRCS the first REC algorithm with guaranteed optimal runtime for this class of distributions, up to the multiplicative constant $\beta$. This significantly improves upon the previous state-of-the-art method, A* coding (Flamich et al., 2022). Under the same assumptions, we experimentally observe and conjecture that the expected runtime and codelength of GRCD are upper bounded by $D_{\mathrm{KL}}[Q\|P] + \mathcal{O}(1)$. Finally, we evaluate GRC in a variational autoencoder-based compression pipeline on MNIST, and show that a modified ELBO and an index-compression method can further improve compression efficiency.

## 1 Introduction and motivation

Over the past decade, the development of excellent deep generative models (DGMs) such as variational autoencoders (VAEs; Vahdat & Kautz, 2020; Child, 2020), normalising flows (Kingma et al., 2016) and diffusion models (Ho et al., 2020) demonstrated great promise in leveraging machine learning (ML) for data compression. Many recent learnt compression approaches have significantly outperformed the best classical hand-crafted codecs across a range of domains, such as lossless and lossy image or video compression (Zhang et al., 2021; Mentzer et al., 2020, 2022).

**Transform coding.** Most learnt compression algorithms are *transform coding* methods: they first map a datum to a latent variable using a learnt transform, and encode it using entropy coding (Ballé et al., 2020). Entropy coding assumes discrete variables while the latent variables in DGMs are typically continuous, so most transform coding methods quantize the latent variable prior to entropy coding. Unfortunately, quantization has zero derivative almost everywhere. Thus, state-of-the-art DGMs trained with gradient-based optimisation must resort to some continuous approximation to

---

[*]Equal contribution.

37th Conference on Neural Information Processing Systems (NeurIPS 2023).

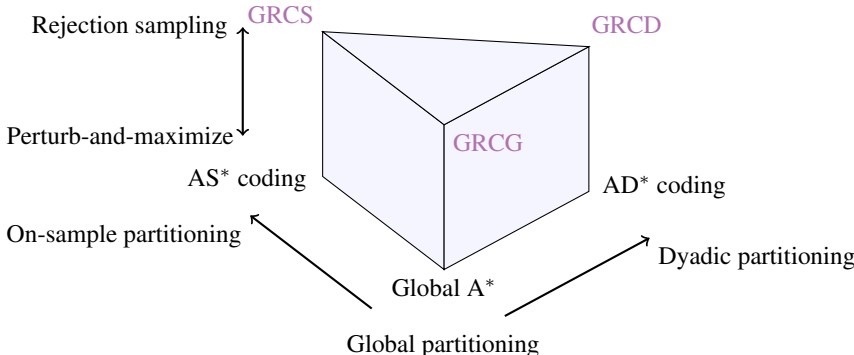

Figure 1: An illustration of the relations between the variants of GRC, introduced in this work, and the variants of A* coding. Algorithms in purple are introduced in this work. The algorithms of Harsha et al. (2007) and Li & El Gamal (2018) are equivalent to GRCG and Global A* coding respectively.

quantisation during training and switch to hard quantisation for compression. Previous works have argued that using quantisation within learnt compression is restrictive or otherwise harmful, and that a method which naturally interfaces with continuous latent variables is needed (Havasi et al., 2018; Flamich et al., 2020; Theis & Agustsson, 2021; Flamich et al., 2022).

**Relative entropy coding.** In this paper, we study an alternative to quantizing and entropy coding the latent representations: we consider randomly perturbing them and encoding the perturbed representations instead. This problem is called relative entropy coding (REC; Havasi et al., 2018; Flamich et al., 2020), and formally, it requires that we encode a random sample from a target $Q$ (the distribution of the perturbed latent representations) using a coding distribution $P$ and some publicly available randomness $S$ (e.g. a publicly shared PRNG seed). Remarkably, there exist algorithms which encode a target sample using only approximately $D_{\mathrm{KL}}[Q\|P]$-many bits on average and, notably, allow $Q$ and $P$ to be continuous (Li & El Gamal, 2018; Flamich et al., 2022; Flamich & Theis, 2023; Flamich, 2023). Thus, we can forgo quantization in our compression pipeline and learn our transforms end-to-end using gradient descent and the reparameterization trick. Moreover, REC has fundamental advantages over quantization in lossy compression with realism constraints (Theis & Agustsson, 2021; Theis et al., 2022) and has a range of other applications, such as differentially private compression for federated learning (Shah et al., 2022; Shahmiri et al., 2023) and compression with Bayesian implicit neural representations (Guo et al., 2023; He et al., 2023).

**Limitations of existing REC algorithms.** While algorithms for solving REC problems already exist, most of them suffer from limitations that render them impractical. These limitations fall into three categories: 1) prohibitively long runtimes, 2) overly restrictive assumptions, or 3) excessive coding overheads. In this work, we study and make progress towards addressing these limitations.

**General-purpose REC algorithms.** The minimal assumption made by any REC algorithm on $Q$ and $P$ is that $D_{\mathrm{KL}}[Q\|P] < \infty$, which ensures that it produces a finite code. However, unfortunately, Agustsson & Theis (2020) showed that without additional assumptions on $Q$ and $P$, the expected runtime of any algorithm that simulates a $Q$-distributed sample using $P$ must scale at least exponentially in $D_{\mathrm{KL}}[Q\|P]$ in the worst case, which is impractically slow for most practical problems. This result holds even in the approximate sense, i.e. the exponential runtime persists even if the algorithm simulates a $\tilde{Q}$-distributed sample with $\|Q - \tilde{Q}\|_{TV} < 1/12$.

**Faster algorithms with additional assumptions.** On the other hand, there exist algorithms which make additional assumptions in order to achieve faster runtimes. For example, dithered quantization (Ziv, 1985; Agustsson & Theis, 2020) achieves an expected runtime of $D_{\mathrm{KL}}[Q\|P]$, which is optimal since any REC algorithm has an expected runtime of at least $D_{\mathrm{KL}}[Q\|P]$. However, it requires both $Q$ and $P$ to be uniform distributions, which limits its applicability. Recently, Flamich et al. (2022) introduced A* coding, an algorithm based on A* sampling (Maddison et al., 2014) which, under assumptions satisfied in practice, achieves an expected runtime of $D_{\infty}[Q\|P]$. Unfortunately, this runtime is suboptimal and is not always practically fast, since $D_{\infty}[Q\|P]$ can be arbitrarily large for fixed $D_{\mathrm{KL}}[Q\|P]$. Further, as discussed in Flamich et al. (2022) this runtime also comes at a cost of an additional, substantial, overhead in codelength, which limits the applicability of A* coding.

**Our contributions.** In this work, we address some of these limitations. First, we propose *greedy rejection coding* (GRC), a REC algorithm based on rejection sampling. Then, inspired by A* coding (Flamich et al., 2022), we develop GRCS and GRCD, two variants of GRC that partition the sample space to dramatically speed up termination. Figure 1 illustrates the relations between GRC and its variants with existing algorithms. We analyze the correctness and the runtime of these algorithms and, in particular, prove that GRCS has an optimal codelength and order-optimal runtime on a wide class of one-dimensional problems. In more detail, our contributions are

- We introduce Greedy Rejection Coding (GRC), which generalises the algorithm of Harsha et al. (2007) to arbitrary probability spaces and partitioning schemes. We prove that under mild conditions, GRC terminates almost surely and returns an unbiased sample from $Q$.

- We introduce GRCS and GRCD, two variants of GRC for continuous distributions over $\mathbb{R}$, which adaptively partition the sample space to dramatically improve their convergence, inspired by AS* and AD* coding (Flamich et al., 2022), respectively.

- We prove that whenever $dQ/dP$ is unimodal, the expected runtime and codelength of GRCS is $\mathcal{O}(D_{\mathrm{KL}}[Q\|P])$. This significantly improves upon the $\mathcal{O}(D_\infty[Q\|P])$ runtime of AS* coding, which is always larger than that of GRCS. This runtime is order-optimal, while making far milder assumptions than, for example, dithered quantization.

- We provide clear experimental evidence for and conjecture that whenever $dQ/dP$ is unimodal, the expected runtime and codelength of GRCD are $D_{\mathrm{KL}}[Q\|P]$. This also significantly improves over the $D_\infty[Q\|P]$ empirically observed runtime of AD* coding.

- We implement a compression pipeline with VAEs, using GRC to compress MNIST images. We propose a modified ELBO objective and show that this, together with a practical method for compressing the indices returned by GRC further improve compression efficiency.

## 2 Background and related work

**Relative entropy coding.** First, we define REC algorithms. Definition 1 is stricter than the one given by Flamich et al. (2022), as it has a stronger condition on the the expected codelength of the algorithm. In this paper, all logarithms are base 2, and all divergences are measured in bits.

**Definition 1** (REC algorithm). *Let $(\mathcal{X}, \Sigma)$ be a measurable space, let $\mathcal{R}$ be a set of pairs of distributions $(Q, P)$ over $(\mathcal{X}, \Sigma)$ such that $D_{\mathrm{KL}}[Q\|P] < \infty$ and $\mathcal{P}$ be the set of all distributions $P$ such that $(Q, P) \in \mathcal{R}$ for some distribution $Q$. Let $S = (S_1, S_2, \dots)$ be a publicly available sequence of independent and fair coin tosses, with corresponding probability space $(\mathcal{S}, \mathcal{F}, \mathbb{P})$ and let $\mathcal{C} = \{0, 1\}^*$ be the set of all finite binary sequences. A REC algorithm is a pair of functions* $\mathrm{enc} : \mathcal{R} \times \mathcal{S} \to \mathcal{C}$ *and* $\mathrm{dec} : \mathcal{C} \times \mathcal{P} \times \mathcal{S} \to \mathcal{X}$, *such that for each $(Q, P) \in \mathcal{R}$ and $S \sim \mathbb{P}$, the outputs of the encoder $C = \mathrm{enc}(Q, P, S)$ and the decoder $X = \mathrm{dec}(P, C, S)$ satisfy*

$$X \sim Q \quad and \quad \mathbb{E}_S[|C|] = D_{\mathrm{KL}}[Q\|P] + \mathcal{O}(\log(D_{\mathrm{KL}}[Q\|P] + 1)), \tag{1}$$

*where $|C|$ is the length of the string $C$. We call* $\mathrm{enc}$ *the encoder and* $\mathrm{dec}$ *the decoder.*

In practice, $S$ is implemented with a pseudo-random number generator (PRNG) with a public seed. In the remainder of this section, we discuss relevant REC algorithms, building up to GRC in section 3.

**Existing REC algorithms.** While there are many REC algorithms already, they suffer from various issues limiting their applicability in practice. Our proposed algorithm, Greedy Rejection Coding (GRC), is based on and generalises Harsha et al. (2007)'s REJ-SAMPLER, by drawing inspiration from A* coding (Flamich et al., 2022). Specifically, A* coding generalises Li & El Gamal (2018)'s *Poisson functional representation* by introducing a partitioning scheme to speed up the algorithm's termination. In an analogous fashion, GRC generalises Harsha et al. (2007) by also introducing partitioning schemes to speed up termination and achieve optimal runtimes. Here we discuss relevant algorithms, building up to GRC in section 3.

**REC with rejection sampling.** In this work, we generalize the rejection sampler introduced by Harsha et al. (2007). While they presented the algorithm for discrete $Q$ and $P$ originally, we generalise it to arbitrary probability spaces in this section and further extend it to arbitrary partitioning schemes (see definition 5) in section 3. The generalisation to arbitrary probability spaces relies on

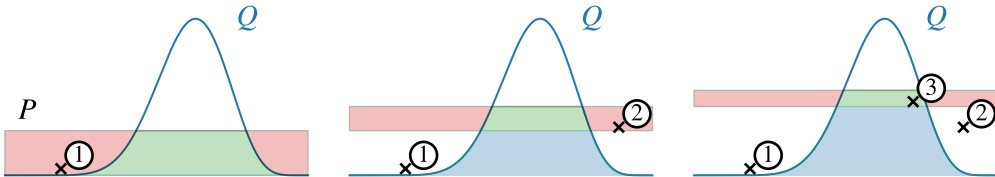

Figure 2: Example run of Harsha et al. (2007), for a pair of continuous $Q$ and $P$ over $[0, 1]$. The green and red regions correspond to acceptance and rejection regions at each step. Here the algorithm rejects the first two samples and accepts the third one, terminating at the third step.

the Radon-Nikodym derivative $dQ/dP$, which is guaranteed to exist since $Q \ll P$ by definition 1. When $Q$ and $P$ both have densities, $dQ/dP$ coincides with the density ratio.

At each step, the algorithm draws a sample from $P$ and performs an accept-reject step, as illustrated in fig. 2. If it rejects the sample, it rules out part of $Q$ corresponding to the acceptance region, adjusts the proposal to account for the removed mass, and repeats until acceptance. More formally, define $T_0$ to be the zero-measure on $\mathcal{X}$, and recursively for $d \in \mathbb{N}$, set:

$$T_{d+1}(S) \stackrel{\text{def}}{=} T_d(S) + A_{d+1}(S), \qquad A_{d+1}(S) \stackrel{\text{def}}{=} \int_S \alpha_{d+1}(x) \, dP(x), \tag{2}$$

$$t_d(x) \stackrel{\text{def}}{=} \frac{dT_d}{dP}(x), \qquad \alpha_{d+1}(x) \stackrel{\text{def}}{=} \min\left\{ \frac{dQ}{dP}(x) - t_d(x), (1 - T_d(\mathcal{X})) \right\}, \tag{3}$$

$$X_d \sim P, \; U_d \sim \text{Uniform}(0, 1) \qquad \beta_{d+1}(x) \stackrel{\text{def}}{=} \frac{\alpha_{d+1}(x)}{1 - T_d(\mathcal{X})}, \tag{4}$$

for all $x \in \mathcal{X}, S \in \Sigma$. The algorithm terminates at the first occurrence of $U_d \leq \beta_{d+1}(X_d)$. The $T_d$ measure corresponds to the mass that has been ruled off up to and including the $d^{\text{th}}$ rejection: $T_1(\mathcal{X}), T_2(\mathcal{X})$ and $T_3(\mathcal{X})$ are the sums of the blue and green masses in the left, middle and right plots of fig. 2 respectively. The $A_d$ measure corresponds to the acceptance mass at the $d^{\text{th}}$ step: $A_1(\mathcal{X}), A_2(\mathcal{X})$ and $A_3(\mathcal{X})$ are the masses of the green regions in the left, middle and right plots of fig. 2 respectively. Lastly, $t_d, \alpha_d$ are the Radon-Nikodym derivatives i.e., roughly speaking, the densities, of $T_d, A_d$ with respect to $P$, and $\beta_{d+1}(X_d)$ is the probability of accepting the sample $X_d$.

To encode the accepted sample $X$, enc outputs the number of rejections $C$ that occurred before acceptance. To decode $X$ from $C$, dec draws $C + 1$ samples from $P$, using the same PRNG seed as the encoder, and returns the last sample. While this algorithm is elegantly simple and achieves optimal codelengths, Flamich & Theis (2023) showed its expected runtime is $2^{D_\infty[Q\|P]}$, where $D_\infty[Q\|P] = \sup_{x \in \mathcal{X}} \log(dQ/dP)(x)$ is the Rényi $\infty$-divergence. Unfortunately, this runtime is prohibitively slow in most practical cases.

**REC with Poisson & Gumbel processes.** Li & El Gamal (2018) introduced a REC algorithm based on Poisson processes, referred to as Poisson Functional Representation (PFR). PFR assumes that $dQ/dP$ is bounded above, and relies on the fact that (Kingman, 1992), if $T_n$ are the ordered arrival times of a homogeneous Poisson process on $\mathbb{R}^+$ and $X_n \sim P$, then

$$N \stackrel{\text{def}}{=} \arg\min_{n \in \mathbb{N}} \left\{ T_n \frac{dP}{dQ}(X_n) \right\} \implies X_N \sim Q. \tag{5}$$

Therefore, PFR casts the REC problem into an optimisation, or search, problem, which can be solved in finite time almost surely. The PFR encoder draws pairs of samples $T_n, X_n$, until it solves the search problem in eq. (5), and returns $X = X_N, C = N - 1$. The decoder can recover $X_N$ from $(P, C, S)$, by drawing $N$ samples from $P$, using the same random seed, and keeping the last sample. While, like the algorithm of Harsha et al. (2007), PFR is elegantly simple and achieves optimal codelengths, its expected runtime is also $2^{D_\infty[Q\|P]}$ (Maddison, 2016).

**Fast REC requires additional assumptions.** These algorithms' slow runtimes are perhaps unsurprising considering Agustsson & Theis's result, which shows under the computational hardness assumption RP $\neq$ NP that without making additional assumptions on $Q$ and $P$, there is no REC

| **Algorithm 1** Harsha et al.'s rejection algorithm; equivalent to GRC with a global partition | **Algorithm 2** GRC with partition process $Z$; differences to Harsha et al.'s algorithm shown in green |
|---|---|
| **Require:** Target $Q$, proposal $P$, space $\mathcal{X}$ | **Require:** Target $Q$, proposal $P$, space $\mathcal{X}$, partition $Z$ |
| 1: $d \leftarrow 0, T_0 \leftarrow 0$ | 1: $d \leftarrow 0, T_0 \leftarrow 0$ |
| 2: | 2: $I_0 \leftarrow 1, S_1 \leftarrow \mathcal{X}$ |
| 3: **while** True **do** | 3: **while** True **do** |
| 4: $\quad X_{d+1} \sim P$ | 4: $\quad X_{I_d} \sim P\|_{S_d}/P(S_d)$ |
| 5: $\quad U_{d+1} \sim \text{Uniform}(0,1)$ | 5: $\quad U_{I_d} \sim \text{Uniform}(0,1)$ |
| 6: $\quad \beta_{d+1} \leftarrow \text{AcceptProb}(Q,P,X_{d+1},T_d)$ | 6: $\quad \beta_{I_d} \leftarrow \text{AcceptProb}(Q,P,X_{I_d},T_d)$ |
| 7: $\quad$ **if** $U_{d+1} \leq \beta_{d+1}$ **then** | 7: $\quad$ **if** $U_{I_d} \leq \beta_{d+1}$ or $d = D_{\max}$ **then** |
| 8: $\quad\quad$ **return** $X_{d+1}, d$ | 8: $\quad\quad$ **return** $X_{I_d}, I_d$ |
| 9: $\quad$ **end if** | 9: $\quad$ **end if** |
| 10: | 10: $\quad p \leftarrow \text{PartitionProb}(Q,P,T_d,Z_{2d},Z_{2d+1})$ |
| 11: | 11: $\quad b_d \sim \text{Bernoulli}(p)$ |
| 12: | 12: $\quad I_{d+1} \leftarrow 2I_d + b_d$ and $S_{d+1} \leftarrow Z_{I_{d+1}}$ |
| 13: $\quad T_{d+1} \leftarrow \text{RuledOutMass}(Q,P,T_d)$ | 13: $\quad T_{d+1} \leftarrow \text{RuledOutMass}(Q,P,T_d,S_{d+1})$ |
| 14: $\quad d \leftarrow d+1$ | 14: $\quad d \leftarrow d+1$ |
| 15: **end while** | 15: **end while** |

algorithm whose expected runtime scales *polynomially* in $D_{\text{KL}}[Q\|P]$. Therefore, in order achieve faster runtimes, a REC algorithm must make additional assumptions on $Q$ and $P$.

**A\* coding.** To this end, Flamich et al. (2022) proposed: (1) a set of appropriate assumptions which are satisfied by many deep latent variable models in practice and (2) a REC algorithm, referred to as A\* coding, which leverages these assumptions to achieve a substantial speed-up over existing methods. In particular, A\* coding generalizes PFR by introducing a partitioning scheme, which splits the sample space $\mathcal{X}$ in nested partitioning subsets, to speed up the solution of eq. (5). Drawing inspiration from this, our proposed algorithm generalises eqs. (2) to (4) in an analogous manner (see fig. 1), introducing partitioning processes (definition 2) to speed up the algorithm's termination.

**Definition 2** (Partitioning process). *A partitioning process is a process* $Z : \mathbb{N}^+ \to \Sigma$ *such that*

$$Z_1 = \mathcal{X}, \;\; Z_{2n} \cap Z_{2n+1} = \emptyset, \;\; Z_{2n} \cup Z_{2n+1} = Z_n. \tag{6}$$

In other words, a partitioning process $Z$ is an infinite binary tree-structured process, where the root node is $Z_1 = \mathcal{X}$ and has index 1, and each node $Z_n$ with index $n$ is partitioned by its two children nodes $Z_{2n}, Z_{2n+1}$ with indices $2n$ and $2n+1$. We refer to this system of indexing as *heap indexing*. In section 3 we present specific choices of partitioning processes which dramatically speed up GRC.

**Greedy Poisson Rejection Sampling.** Contemporary to our work, Flamich (2023) introduces a rejection sampler based on Poisson processes, called Greedy Poisson Rejection Sampling (GPRS), which can be used as a REC algorithm. Similar to GRC and A\* coding, GPRS partitions the sample space to speed up the convergence to the accepted sample. Furthermore, a variant of GPRS also achieves order-optimal runtime for one-dimensional distribution pairs with a unimodal density ratio. However, the construction of their method is significantly different from ours, relying entirely on Poisson processes. Moreover, GPRS requires numerically solving a certain ODE, while our method does not, making it potentially more favourable in practice. We believe establishing a closer connection between GPRS and GRC is a promising future research direction.

## 3 Greedy Rejection Coding

**Generalising Harsha et al. (2007).** In this section we introduce Greedy Rejection Coding (GRC; definition 5), which generalises the algorithm of Harsha et al. (2007) in two ways. First, GRC can be used with distributions over arbitrary probability spaces. Therefore, it is applicable to arbitrary REC problems, including REC with continuous distributions. Second, similar to A\* coding, GRC can be combined with arbitrary partitioning processes, allowing it to achieve optimal runtimes given additional assumptions on the REC problem, and an appropriate choice of partitioning process.

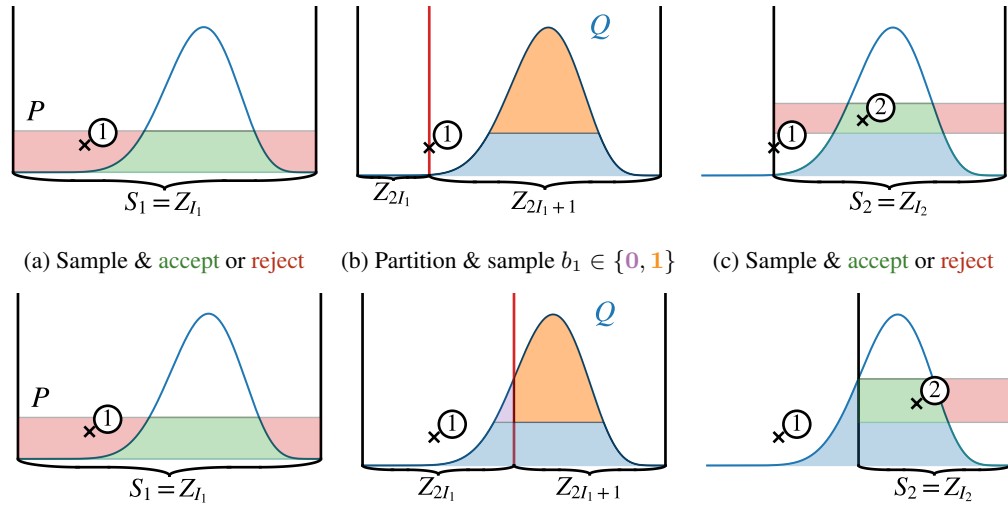

(a) Sample & accept or reject    (b) Partition & sample $b_1 \in \{0, 1\}$    (c) Sample & accept or reject

(d) Sample & accept or reject    (e) Partition & sample $b_1 \in \{0, 1\}$    (f) Sample & accept or reject

Figure 3: Illustrations of the two variants of GRC considered in this work. (a) to (c) show GRC with the *on-sample* partitioning process (GRCS). (d) to (f) show GRC with the dyadic partition process (GRCD). GRC interleaves accept-reject steps with partitioning steps. In the former, it draws a sample and either accepts or rejects it. In the latter, it partitions the sample space and randomly chooses one of the partitions, ruling out large parts of the sample space and speeding up termination.

## 3.1 Algorithm definition

**Overview.** Before specifying GRC, we summarise its operation with an accompanying illustration. On a high level, GRC interleaves accept-reject steps with partitioning steps, where the latter are determined by a partitioning process. Specifically, consider the example in figs. 3d to 3f, where $Q$ and $P$ are distributions over $\mathcal{X} = [0, 1]$, and $Z$ is the partitioning process defined by

$$Z_n = [L, R] \implies Z_{2n} = [L, M], Z_{2n+1} = [M, R], \text{ where } M = (L + R)/2. \tag{7}$$

In each step $d = 1, 2, \ldots$, GRC maintains a heap index $I_d$ of an infinite binary tree, and an active subset $S_d = Z_{I_d} \subseteq \mathcal{X}$ of the sample space, initialised as $I_0 = 1$ and $S_1 = Z_1 = \mathcal{X}$ respectively.

**Accept-reject step.** In each step, GRC draws a sample from the restriction of $P$ to $S_d$, namely $P|_{S_d}/P(S_d)$, and either accepts or rejects it. If the sample is accepted, the algorithm terminates. Otherwise, GRC performs a partitioning step as shown in fig. 3d.

**Partitioning step.** In each partitioning step, GRC partitions $S_d = Z_{I_d}$ into $Z_{2I_d}$ and $Z_{2I_d+1}$, as specified by the partitioning process $Z$. It then samples a Bernoulli random variable $b_d$, whose outcomes have probabilities proportional to the mass of $Q$ which has not been accounted for, up to and including step $d$, within the partitions $Z_{2I_d}$ and $Z_{2I_d+1}$ respectively. In fig. 3e, these two masses correspond to the purple and orange areas, and the algorithm has sampled $b_d = 1$. Last, GRC updates the heap index to $I_{d+1} = 2I_d + b_d$ and the active subset to $S_{d+1} = Z_{I_{d+1}}$. GRC proceeds by interleaving accept-reject and partitioning steps until an acceptance occurs.

**Algorithm specification.** The aforementioned algorithm can be formalised in terms of probability measures over arbitrary spaces and arbitrary partitioning processes. Above, algorithms 1 and 2 describe Harsha et al.'s rejection sampler and our generalisation of it, respectively. For the sake of keeping the exposition lightweight, we defer the formal measure-theoretic definition of GRC to the appendix (see definition 5 in appendix A.1), and refer to algorithm 2 as a working definition here.

**Comparison to Harsha et al.** While algorithms 1 and 2 are similar, they differ in two notable ways. First, rather than drawing a sample from $P$, GRC draws a sample from the restriction of $P$ to an active subset $S_d = Z_d \subseteq \mathcal{X}$, namely $P|_{S_d}/P(S_d)$. Second, GRC updates its active subset $S_d = Z_d$ at each step, setting it to one of the children of $Z_d$, namely either $Z_{2d}$ or $Z_{2d+1}$, by drawing $b_d \sim$ Bernoulli, and setting $Z_{2d+b_d}$. This partitioning mechanism, which does not appear in algorithm 1, yields a

different variant of GRC for each choice of partitioning process $Z$. In fact, as shown in Proposition 1 below, algorithm 1 is a special case of GRC with $S_d = \mathcal{X}$ for all $d$. See appendix A.2 for the proof.

**Proposition 1** (Harsha et al. (2007) is a special case of GRC). *Let $Z$ be the global partitioning process over $\Sigma$, defined as*

$$Z_1 = \mathcal{X}, \quad Z_{2n} = Z_n, \quad Z_{2n+1} = \emptyset, \quad for\ all\ \ n = 1, 2, \dots. \tag{8}$$

*Harsha et al. (2007) is equivalent to GRC using this $Z$ and setting $C = D^*$ instead of $C = I_{D^*}$. We refer to this algorithm as Global GRC, or **GRCG** for short.*

**Partitioning processes and additional assumptions.** While Proposition 1 shows that Harsha et al.'s algorithm is equivalent to GRC with a particular choice of $Z$, a range of other choices of $Z$ is possible, and this is where we can leverage additional structure. In particular, we show that when $Q$ and $P$ are continuous distributions over $\mathbb{R}$ with a unimodal density ratio $dQ/dP$, we can dramatically speed up GRC with an appropriate choice of $Z$. In particular, we will consider the on-sample and dyadic partitioning processes from Flamich et al. (2022), given in Definitions 3 and 4.

**Definition 3** (On-sample partitioning process). *Let $\mathcal{X} = \mathbb{R} \cup \{-\infty, \infty\}$ and $P$ a continuous distribution. The on-sample partitioning process is defined as*

$$Z_n = [a, b], a, b \in \mathcal{X} \implies Z_{2n} = [a, X_n], \quad Z_{2n+1} = [X_n, b], \ where\ X_n \sim P|_{Z_n}/P(Z_n).$$

In other words, in the on-sample partitioning process, $Z_n$ are intervals of $\mathbb{R}$, each of which is partitioned into sub-intervals $Z_{2n}$ and $Z_{2n+1}$ by splitting at the sample $X_n$ drawn from $P|_{Z_n}/P(Z_n)$. We refer to GRC with on-sample partitioning as **GRCS**.

**Definition 4** (Dyadic partitioning process). *Let $\mathcal{X} = \mathbb{R} \cup \{-\infty, \infty\}$ and $P$ a continuous distribution. The dyadic partitioning process is defined as*

$$Z_n = [a, b], a, b \in \mathcal{X} \implies Z_{2n} = [a, c], \quad Z_{2n+1} = [c, b], \ such\ that\ P(Z_{2n}) = P(Z_{2n+1}).$$

Similar to on-sample partitioning, in the dyadic process $Z_n$ are intervals of $\mathbb{R}$. However, in the dyadic process, $Z_n$ is partitioned into sub-intervals $Z_{2n}$ and $Z_{2n+1}$ such that $P(Z_{2n}) = P(Z_{2n+1})$. We refer to GRC with the dyadic partitioning process as **GRCD**.

**GRC with a tunable codelength.** Flamich et al. (2022) presented a depth-limited variant of AD* coding, DAD* coding, in which the codelength $|C|$ can be provided as a tunable input to the algorithm. Fixed-codelength REC algorithms are typically approximate because they introduce bias in their samples, but are nevertheless useful in certain contexts, such as for coding a group of random variables with the same fixed codelength. GRCD can be similarly modified to accept $|C|$ as an input, by limiting the maximum steps of the algorithm by $D_{\max}$ (see algorithm 2). Setting $D_{\max} = \infty$ in algorithm 2 corresponds to exact GRC, while setting $D_{\max} < \infty$ corresponds to depth-limited GRC. In appendix D we provide detailed analysis of depth-limited GRC.

## 3.2 Theoretical results

**Correctness of GRC.** Below, we present three sets of assumptions on $Q, P$ and $Z$, and in theorem 1, we show that fulfilling any of them is sufficient to ensure the correctness of GRC.

**Assumption 1.** *GRC has a finite ratio mode, i.e. $dQ/dP(x) < M$ for all $x \in \mathcal{X}$, for some $M > 0$.*

Assumption 1 is the most generally applicable, as it does not restrict the sample space. Assumption 1 holds for GRCG, GRCS and GRCD, so long as $dQ/dP$ is bounded. While this assumption is very general, in some cases we may want to consider $Q, P$ with unbounded $dQ/dP$. To this end, we show that it can be replaced by alternative assumptions, such as assumptions 2 and 3.

**Assumption 2.** *GRC is single-branch, i.e. for each $d$, $b_d = 0$ or $b_d = 1$ almost surely.*

GRC with the global partitioning process (eq. 8) satisfies assumption 2. In addition, if $Q$ and $P$ are distributions over $\mathbb{R}$ and $dQ/dP$ is unimodal, GRCS also satisfies assumption 2.

**Assumption 3.** *$\mathcal{X} \subseteq \mathbb{R}^N$ and GRC has nicely-shrinking bounds, i.e. $\mathbb{P}$-almost surely the following holds: for each $x \in \mathcal{X}$ which is in a nested sequence of partitions $x \in Z_1 \supseteq \dots \supseteq Z_{k_d} \supseteq \dots$ with $P(Z_{k_d}) \to 0$ as $k_d \to \infty$, there exist $\gamma, r_1, r_2, \dots \in \mathbb{R}_{>0}$ such that*

$$r_d \to 0, \ Z_{k_d} \subseteq B_{r_d}(x) \ and\ P(Z_{k_d}) \geq \gamma P(B_{r_d}(x)), \tag{9}$$

*where $B_r(x)$ denotes the open ball of radius $r$ centered on $x$. We recall that $\mathbb{P}$ is the measure associated with the sequence of publicly available coin flips $S$.*

Intuitively, $Z$ is nicely shrinking if its branches shrink in a roughly uniform way over $\mathcal{X}$. As the most important example, when $Q$ and $P$ are distributions over $\mathbb{R}$, GRCD satisfies assumption 3. Now, Theorem 1 shows that if any of the above assumptions hold, then GRC terminates almost surely and yields unbiased samples from $Q$. We provide the proof in appendix B.

**Theorem 1** (Correctness of GRC). *Suppose $Q, P$ and $Z$ satisfy any one of assumptions 1 to 3. Then, algorithm 2 terminates with probability 1, and its returned sample $X$ has law $X \sim Q$.*

**Expected runtime and codelength of GRCS.** Now we turn to the expected runtime and codelength of GRCS. Theorem 2 shows that the expected codelength of GRCS is optimal, while Theorem 3 establishes that its runtime is order-optimal. We present the proofs of the theorems in appendix C.

**Theorem 2** (GRCS codelength). *Let $Q$ and $P$ be distributions over $\mathbb{R}$ with $D_{\mathrm{KL}}[Q\|P] < \infty$ and $dQ/dP$ unimodal. Let $Z$ be the on-sample partitioning process, and $X$ its returned sample. Then,*

$$\mathbb{H}[X|Z] \leq D_{\mathrm{KL}}[Q\|P] + 2\log\left(D_{\mathrm{KL}}[Q\|P] + 1\right) + \mathcal{O}(1). \tag{10}$$

**Theorem 3** (GRCS runtime). *Let $Q$ and $P$ be distributions over $\mathbb{R}$ with $D_{\mathrm{KL}}[Q\|P] < \infty$ and $dQ/dP$ unimodal. Let $Z$ be the on-sample partitioning process and $D$ the number of steps the algorithm takes before accepting a sample. Then, for $\beta = 2/\log(4/3) \approx 4.82$ we have*

$$\mathbb{E}[D] \leq \beta\, D_{\mathrm{KL}}[Q\|P] + \mathcal{O}(1). \tag{11}$$

**Improving the codelength of GRCD.** In theorem 2, we state the bound for the REC setting, where we make no further assumptions on $Q$ and $P$. However, we can improve the bound if we consider the *reverse channel coding* (RCC) setting (Theis & Yosri, 2022). In RCC, we have a pair of correlated random random variables $X, Y \sim P_{X,Y}$. During one round of communication, the encoder receives $Y \sim P_Y$ and needs to encode a sample $X \sim P_{X|Y}$ from the posterior using $P_X$ as the proposal distribution. Thus, RCC can be thought of as the average-case version of REC, where the encoder sets $Q \leftarrow P_{X|Y}$ and $P \leftarrow P_X$. In this case, when the conditions of theorem 2 hold for every $(P_{X|Y}, P_X)$ pair, in appendix C we show that the coefficient of the $\log$-factor in eq. (10) can be improved so that the average-case bound becomes $\mathbb{I}[X;Y] + \log(\mathbb{I}[X;Y] + 1) + \mathcal{O}(1)$, where $\mathbb{I}[X;Y] = \mathbb{E}_{Y \sim P_Y}\left[D_{\mathrm{KL}}[P_{X|Y}\|P_Y]\right]$ is the mutual information between $X$ and $Y$.

**GRCS runtime is order-optimal.** Theorem 3 substantially improves upon the runtime of $A^*$ coding, which is the current fastest REC algorithm with similar assumptions. In particular, $AS^*$ coding has $\mathcal{O}(D_\infty[Q\|P])$ expected runtime, which can be arbitrarily larger than that of GRCS. Remarkably, the runtime of GRCS is optimal up to the multiplicative factor $\beta$.

## 4 Experiments

We conducted two sets of experiments: one on controlled synthetic REC problems to check the predictions of our theorems numerically, and another using VAEs trained on MNIST to study how the performance of GRC-based compression pipelines can be improved in practice. We conducted all our experiments under fair and reproducible conditions. Our code is available at https://github.com/cambridge-mlg/fast-rec-with-grc.

### 4.1 Synthetic Experiments

**Synthetic REC experiments.** First, we compare GRCS and GRCD, against $AS^*$ and $AD^*$ coding, on a range of synthetic REC problems. We systematically vary distribution parameters to adjust the difficulty of the REC problems. Figure 4 shows the results of our synthetic experiments.

**Partitioning processes improve the runtime of GRC.** First, we observe that, assuming that $dQ/dP$ is unimodal, introducing the on-sample or the dyadic partitioning process speeds up GRC dramatically. In particular, fig. 4 shows that increasing the infinity divergence $D_\infty[Q\|P]$ (for a fixed $D_{\mathrm{KL}}[Q\|P]$) does not affect the runtimes of GRCS and GRCD, which remain constant and small. This is a remarkable speed-up over the exponential expected runtime of GRCG.

**GRC is faster than $A^*$ coding.** Further, we observe that GRC significantly improves upon the runtime of A* coding, which is the fastest previously known algorithm with similar assumptions. In particular, Figure 4 shows that increasing the infinity divergence $D_\infty[Q\|P]$, while keeping the

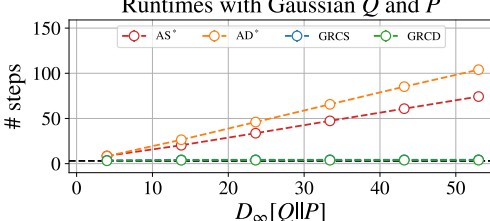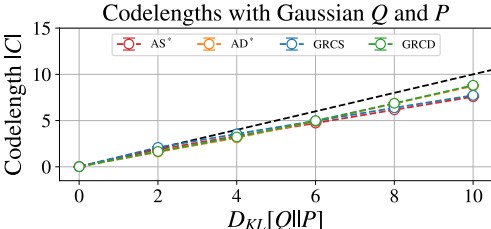

Figure 4: Comparison between GRC and A* coding on synthetic REC problems with Gaussian $Q$ and $P$. *Left:* we fix $D_{\mathrm{KL}}[Q\|P] = 3$ and vary $D_\infty[Q\|P]$, measuring the number of steps taken by each algorithm. *Right:* we fix $D_\infty[Q\|P] = D_{\mathrm{KL}}[Q\|P] + 2$ and vary $D_{\mathrm{KL}}[Q\|P]$, plotting the codelengths produced by each algorithm. Reported codelengths do not include additional logarithmic overhead terms. Results are averaged over $4 \times 10^3$ different random seeds for each datapoint. We have included error-bars in both plots but these are too small to see compared to the plot scales.

KL divergence $D_{\mathrm{KL}}[Q\|P]$ fixed, increases the runtime of both AS* and AD* coding, while the runtimes of GRCS and GRCD remain constant. More generally, for a fixed KL divergence, the infinity divergence can be arbitrarily large or even infinite. In such cases, A* coding would be impractically slow or even inapplicable, while GRCS and GRCD remain fast.

### 4.2 Compression with Variational Autoencoders

**Compressing images with VAEs and REC.** One of the most promising applications of REC is in learnt compression. Here, we implement a proof-of-concept lossless neural compression pipeline using a VAE with a factorized Gaussian posterior on MNIST and take the architecture used by [Townsend et al. (2018)](). To compress an image $Y$, we encode a latent sample $X$ from the VAE posterior $q(X \mid Y)$ by applying GRCD dimensionwise after which we encode the image $Y$ with entropy coding using the VAE's conditional likelihood $p(Y \mid X)$ as the coding distribution. Unfortunately, in addition to the $D_{\mathrm{KL}}[q(X_d \mid Y)\|p(X_d)]$ bits coding cost for latent dimension $d$, this incurs an overhead of $\log(D_{\mathrm{KL}}[q(X_d \mid Y)\|p(X_d)] + 1) + \mathcal{O}(1)$ bits, analogously to how a symbol code, like Huffman coding, incurs a constant overhead per symbol ([MacKay, 2003]()). However, since $\log(1 + x) \approx x$ when $x \approx 0$, the logarithmic overhead of GRC can become significant compared to the KL divergence. Hence, we now investigate two approaches to mitigate this issue.

**Modified ELBO for REC.** A principled approach to optimizing our neural compression pipeline is to minimize its expected codelength. For bits-back methods ([Townsend et al., 2018, 2019]()), the negative ELBO indeed expresses their expected codelength, but in REC's case, it does not take into account the additional dimensionwise logarithmic overhead we discussed above. Thus, we propose to minimize a modified negative ELBO to account for this (assuming that we have $D$ latent dimensions):

$$\underbrace{\mathbb{E}_{X\sim q(X|Y)}[-\log p(Y|X)] + D_{\mathrm{KL}}[q(X|Y)\|p(X)]}_{\text{Regular ELBO}} + \sum_{d=1}^{D} \underbrace{\log\left(D_{\mathrm{KL}}[q(X_d|Y)\|p(X_d)] + 1\right)}_{\text{Logarithmic overhead per dimension}}. \quad (12)$$

**Coding the latent indices.** As the final step during the encoding process, we need a prefix code to encode the heap indices $I_d$ returned by GRCD for each $d$. Without any further information, the best we can do is use Elias $\delta$ coding ([Elias, 1975]()), which, assuming our conjecture on the expected runtime of GRCD holds, yields an expected codelength of $\mathbb{I}[Y; X] + 2\log(\mathbb{I}[Y; X] + 1) + \mathcal{O}(1)$. However, we can improve this if we can estimate $\mathbb{E}[\log I_d]$ for each $d$: it can be shown, that the maximum entropy distribution of a positive integer-valued random variable with under a constraint on the expectation on its logarithm is $\zeta(n|\lambda) \propto n^{-\lambda}$, with $\lambda^{-1} = \mathbb{E}[\log I_d] + 1$. In this case, entropy coding $I_d$ using this $\zeta$ distribution yields improves the expected codelength to $\mathbb{I}[Y; X] + \log(\mathbb{I}[Y; X] + 1) + \mathcal{O}(1)$.

**Experimental results.** We trained our VAE with $L \in \{20, 50, 100\}$ latent dimensions optimized using the negative ELBO and its modified version in Equation ([12]()), and experimented with encoding the heap indices of GRCD with both $\delta$ and $\zeta$ coding. We report the results of our in Table [1]() on the MNIST test set in bits per pixel. In addition to the total coding cost, we report the negative ELBO per pixel, which is the fundamental lower bound on the compression efficiency of REC with each

| Training objective | # Latent | Total BPP with $\zeta$ coding | Total BPP with $\delta$ coding | Neg. ELBO per pixel | Overhead BPP with $\delta$ coding |
|---|---|---|---|---|---|
| ELBO | 20 | $1.472 \pm 0.004$ | $1.482 \pm 0.004$ | $1.391 \pm 0.004$ | $0.091 \pm 0.000$ |
| | 50 | $1.511 \pm 0.003$ | $1.530 \pm 0.003$ | $1.357 \pm 0.003$ | $0.172 \pm 0.000$ |
| | 100 | $1.523 \pm 0.003$ | $1.600 \pm 0.003$ | $1.362 \pm 0.003$ | $0.238 \pm 0.000$ |
| Modified ELBO | 20 | $1.470 \pm 0.004$ | $1.478 \pm 0.004$ | $1.393 \pm 0.004$ | $0.085 \pm 0.000$ |
| | 50 | $1.484 \pm 0.003$ | $1.514 \pm 0.003$ | $1.373 \pm 0.003$ | $0.141 \pm 0.000$ |
| | 100 | $1.485 \pm 0.003$ | $1.579 \pm 0.003$ | $1.373 \pm 0.003$ | $0.205 \pm 0.000$ |

Table 1: Lossless compression performance comparison on the MNIST test set of a small VAE with different latent space sizes, optimized using either the ELBO or the modified ELBO in eq. (12). We report the bits per pixel (BPP) attained using different coding methods, averaged over the 10,000 test images, along with the standard error, using GRCD. See section 4.2 for further details.

VAE. Finally, we report the logarithmic overhead due to $\delta$ coding. We find that both the modified ELBO and $\zeta$ coding prove beneficial, especially as the dimensionality of the latent space increases. This is expected, since the overhead is most significant for latent dimensions with small KLs, which becomes more likely as the dimension of the latent space grows. The improvements yielded by each of the two methods are significant, with $\zeta$ coding leading to a consistent $1 - 7\%$ gain compared to $\delta$ coding and the modified objective resulting in up to $2\%$ gain in coding performance.

## 5 Conclusion and Future Work

**Summary.** In this work, we introduced Greedy Rejection Coding (GRC), a REC algorithm which generalises the rejection algorithm of Harsha et al. to arbitrary probability spaces and partitioning processes. We proved the correctness of our algorithm under mild assumptions, and introduced GRCS and GRCD, two variants of GRC. We showed that the runtimes of GRCS and GRCD significantly improve upon the runtime of $A^*$ coding, which can be arbitrarily larger. We evaluated our algorithms empirically, verifying our theory and conducted a proof-of-concept learnt compression experiment on MNIST using VAEs. We demonstrated that a principled modification to the ELBO and entropy coding GRCD's indices using a $\zeta$ distribution can further improve compression efficiency.

**Limitations and Further work.** One limitation of GRC is that, unlike $A^*$ coding, it requires us to be able to evaluate the CDF of $Q$. While in some settings this CDF may be intractable, this assumption is satisfied by most latent variable generative models, and is not restrictive in practice. However, one practical limitation of GRCS and GRCD, as well as AS$^*$ and AD$^*$, is that they assume target-proposal pairs over $\mathbb{R}$. For multivariate distributions, we can decompose them into univariate conditionals and apply GRC dimensionwise, however this incurs an additional coding overhead per dimension, resulting in a non-negligible cost. Thus, an important direction is to investigate whether fast REC algorithms for multivariate distributions can be devised, to circumvent this challenge.

## 6 Author Contributions

SM suggested that single-branch greedy rejection coding (Flamich & Theis, 2023) can be extended to arbitrary partitioning processes. GF discovered GRCS and SM discovered GRCD. SM provided a proof for Theorem 1 and GF provided proofs for Theorems 2 and 3. SM carried out the synthetic experiments and GF carried out the VAE experiments. SM drafted the majority of the paper, while GF wrote section 4.2 and parts of sections 2 and 3.2. They contributed to the editing of the paper equally. JMH supervised and steered the project.

## 7 Acknowledgements

GF acknowledges funding from DeepMind. SM acknowledges funding from the Qualcomm Innovation Fellowship and the Vice Chancellor's & George and Marie Vergottis scholarship of the Cambridge Trust.

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
