# A  Formal definition of Greedy Rejection Coding

## A.1  Formal definition

Here we give a formal definition of GRC in terms of measures. We chose to omit this from the main text for the sake of exposition, and instead formally define GRC in definition 5 below.

**Definition 5** (Greedy Rejection Coding). *Let $Z$ be a partitioning process on $\Sigma$, and $I_0 = 1$, $S_0 = Z_{I_0}$. Let $T_0(\cdot, S_0)$ be the zero-measure on $(\mathcal{X}, \Sigma)$. Then for $d = 0, 1, \dots$ define*

$$t_d(x, S_{0:d}) \stackrel{\text{def}}{=} \frac{dT_d(\cdot, S_{0:d})}{dP(\cdot)}(x), \tag{13}$$

$$\alpha_{d+1}(x, S_{0:d}) \stackrel{\text{def}}{=} \min\left\{ \frac{dQ}{dP}(x) - t_d(x, S_{0:d}), \frac{1 - T_d(\mathcal{X}, S_{0:d})}{P(S_d)} \right\} \tag{14}$$

$$A_{d+1}(S, S_{0:d}) \stackrel{\text{def}}{=} \int_S dP(x) \, \alpha_{d+1}(x, S_{0:d}), \tag{15}$$

$$\beta_{d+1}(x, S_{0:d}) \stackrel{\text{def}}{=} \alpha_{d+1}(x, S_{0:d}) \, \frac{P(S_d)}{1 - T_d(\mathcal{X}, S_{0:d})}, \tag{16}$$

$$X_{I_d} \sim \frac{P|_{S_d}}{P(S_d)}, \tag{17}$$

$$U_{I_d} \sim \text{Uniform}(0, 1), \tag{18}$$

$$b_d \sim \text{Bernoulli}\left( \frac{Q(Z_{2I_d+1}) - T_d(Z_{2I_d+1}, S_{0:d}) - A_{d+1}(Z_{2I_d+1}, S_{0:d})}{Q(S_d) - T_d(S_d, S_{0:d}) - A_{d+1}(S_d, S_{0:d})} \right), \tag{19}$$

$$I_{d+1} \stackrel{\text{def}}{=} 2I_d + b_d, \tag{20}$$

$$S_{d+1} \stackrel{\text{def}}{=} Z_{I_{d+1}}, \tag{21}$$

$$T_{d+1}(S, S_{0:d+1}) \stackrel{\text{def}}{=} T_d(S \cap S_{d+1}, S_{0:d}) + A_{d+1}(S \cap S_{d+1}, S_{0:d}) + Q(S \cap S'_{d+1}), \tag{22}$$

*where $S \in \Sigma$ and $P|_{Z_d}$ denotes the restriction of the measure $P$ to the set $Z_d$. Generalised Greedy Rejection Coding (GRC) amounts to running this recursion, computing*

$$D^* = \min\{d \in \mathbb{N} : U_{I_d} \leq \beta_{d+1}(X_{I_d}, S_{0:d})\}, \tag{23}$$

*and returning $X = X_{I_{D^*}}$ and $C = I_{D^*}$.*

The functions `AcceptProb` and `RuledOutMass` in algorithm 2 correspond to calculating the quantities in eq. (16) and eq. (22). The function `PartitionProb` corresponds to computing the success probability of the Bernoulli coin toss in eq. (19).

## A.2  Harsha et al.'s algorithm is a special case of GRC

Here we show that the algorithm of Harsha et al. is a special case of GRC which assumes discrete $P$ and $Q$ distributions and uses the global partitioning process, which we refer to as GRCG. Note that the original algorithm described by Harsha et al. assumes discrete $P$ and $Q$ distributions, whereas GRCG does not make this assumption.

**Proposition 2** (Harsha et al. (2007) is a special case of GRC). *Let $Z$ be the global partitioning process over $\Sigma$, defined as*

$$Z_1 = \mathcal{X}, \quad Z_{2n} = Z_n, \quad Z_{2n+1} = \emptyset, \quad \text{for all } n = 1, 2, \dots. \tag{24}$$

*Harsha et al. (2007) is equivalent to GRC using this $Z$ and setting $C = D^*$ instead of $C = I_{D^*}$. We refer to this variant of GRC as Global GRC, or GRCG for short.*

*Proof.* With $Z$ defined as in eq. (24), we have $b_d \sim \text{Bernoulli}(0)$ by eq. (19), so $b_d = 0$ almost surely. Therefore $S_d = \mathcal{X}$ for all $d \in \mathbb{N}^+$. From this, we have $T_{d+1}(S, S_{0:d}) = T_d(S, S_{0:d}) + A_d(S, S_{0:d})$ and also $P(S_d) = P(\mathcal{X}) = 1$ for all $d \in \mathbb{N}^+$. Substituting these in the equations of definition 5, we recover eqs. (2) to (4). Setting $C = D^*$ instead of $C = I_{D^*}$ makes the two algorithms identical. $\quad\square$

# B   Proof of correctness of GRC: Theorem 1

In this section we give a proof for the correctness of GRC. Before going into the proof, we outline our approach and the organisation of the proof.

**Proof outline.** To prove theorem 1, we consider running GRC for a finite number of $d$ steps. We consider the measure $\tau_d : \Sigma \to [0, 1]$, defined such that for any $S \in \Sigma$, the quantity $\tau_d(S)$ is equal to the probability that GRC terminates within $d$ steps and returns a sample $X \in S \subseteq \Sigma$. We then show that $\tau_d \to Q$ in total variation as $d \to \infty$, which proves theorem 1.

**Organisation of the proof.** First, in section B.1 we introduce some preliminary definitions, assumptions and notation on partitioning processes, which we will use in later sections. Then, in B.2 we derive the $\tau_d$ measure, and prove some intermediate results about it. Specifically, proposition 3 shows that the measures $A_d$ and $T_d$ from the definition of GRC (definition 5) correspond to probabilities describing the termination of the algorithm, and lemma 1 uses these facts to derive the form of $\tau_d$ in terms of $A_d$. Then, lemma 2 shows that the measure $\tau_d$ is no larger than the measure $Q$ and lemma 3 shows that the limit of $\tau_d$ as $d \to \infty$ is also a measure. Lastly lemma 4 shows that $T_d$ and $\tau_d$ are equal on the active sets of the partition process followed within a run of GRC, and then lemma 5 uses that result to derive the subsets of the sample space on which $\tau_d$ is equal to $Q$ and $\tau$ is equal to $Q$.

Then, in appendix B.3 we break down the proof of theorem 1 in four cases. First, we consider the probability $p_d$ that GRC terminates at step $d$, given that it has not terminated up to and including step $d - 1$. Lemma 7 shows that if $p_d \not\to 0$, then $\tau_d \to Q$ in total variation. Then we consider the case $p_d \to 0$ and show that in this case, if any of assumptions 1, 2 or 3 hold, then again $\tau_d \to Q$ in total variation. Putting these results together proves theorem 1.

## B.1   Preliminary definitions, assumptions and notation

For the sake of completeness, we restate relevant definitions and assumptions. Definition 6 restates our notation on the target $Q$ and proposal $P$ measures and assumption 4 emphasises our assumption that $Q \ll P$. Definition 7 restates the definition of partitioning processes.

**Definition 6** (Target $Q$ and proposal $P$ distributions). *Let $Q$ and $P$ be probability measures on a measurable space $(\mathcal{X}, \Sigma)$. We refer to $Q$ and $P$ as the target and proposal measures respectively.*

**Assumption 4** ($Q \ll P$). *We assume $Q$ is absolutely continuous w.r.t. $P$, that is $Q \ll P$. Under this assumption, the Radon-Nikodym derivative of $Q$ w.r.t. $P$ exists and is denoted as $dQ/dP : \mathcal{X} \to \mathbb{R}^+$.*

**Definition 7** (Partitioning process). *A random process $Z : \mathbb{N}^+ \to \Sigma$ which satisfies*

$$Z_1 = \mathcal{X}, \;\; Z_{2n} \cap Z_{2n+1} = \emptyset, \;\; Z_{2n} \cup Z_{2n+1} = Z_n. \tag{25}$$

*is called a partitioning process.*

That is, a partitioning process $Z$ is a random process indexed by the heap indices of an infinite binary tree, where the root node is $\mathcal{X}$ and any two children nodes $Z_{2n}$ and $Z_{2n+1}$ partition their parent node $Z_n$. Note that by definition, a partitioning process takes values which are measurable sets in $(\mathcal{X}, \Sigma)$.

Because GRC operates on an binary tree, we find it useful to define some appropriate notation. Definition 8 specifies the ancestors of a node in a binary tree. Notation 1 gives some useful indexing notation for denoting different elements of the partitioning process $Z$, as well as for denoting the branch of ancestors of an element in a partitioning process.

**Definition 8** (Ancestors). *We define the one-step ancestor function $A_1 : 2^{\mathbb{N}^+} \to 2^{\mathbb{N}^+}$ as*

$$A_1(N) = N \cup \{n \in \mathbb{N}^+ : n' = 2n \text{ or } n' = 2n + 1, \text{ for some } n' \in N\}, \tag{26}$$

*and the ancestor function $A : 2^{\mathbb{N}^+} \to 2^{\mathbb{N}^+}$ as*

$$A(N) = \left\{n \in \mathbb{N}^+ : n \in A_1^k(\{n'\}) \text{ for some } n' \in N, k \in \mathbb{N}^+\right\}. \tag{27}$$

*where $A_1^k$ denotes the composition of $A_1$ with itself $k$ times.*

Viewing $\mathbb{N}^+$ as the set of heap indices of an infinite binary tree, $A$ maps a set $N \subseteq \mathbb{N}$ of natural numbers (nodes) to the set of all elements of $N$ and their ancestors.

**Notation 1** (Double indexing for $Z$, ancestor branch). *Given a partitioning process $Z$, we use the notation $Z_{d,k}$, where $d = 1, 2, \ldots$ and $k = 1, \ldots, 2^{d-1}$ to denote the $k^{th}$ node at depth $d$, that is*

$$Z_{d,k} := Z_{2^{d-1}-1+k}. \tag{28}$$

*We use the hat notation $\hat{Z}_{d,k}$ to denote the sequence of nodes consisting of $Z_{d,k}$ and all its ancestors*

$$\hat{Z}_{d,k} := (Z_n : n \in A(\{2^{d-1} - 1 + k\})), \tag{29}$$

*and call $\hat{Z}_{d,k}$ the ancestor branch of $Z_{d,k}$.*

**Notation 2** ($\mathbb{P}$ measure). *In definition 5, we defined $\mathbb{P}$ to be the measure associated with an infinite sequence of independent fair coin tosses over a measurable space $(\Omega, \mathcal{S})$. To avoid heavy notation, for the rest of the proof we will overload this symbol as follows: if $F$ is a random variable from $\Omega$ to some measurable space, we will abbreviate $\mathbb{P} \circ F^{-1}$ by simply $\mathbb{P}(F)$.*

## B.2 Deriving the measure of samples returned by GRC

For the remainder of the proof, we condition on a fixed partitioning process sample $Z$. For brevity, we omit this conditioning which, from here on is understood to be implied. Proposition 3 shows that the measures $A_d$ and $T_d$ correspond to the probabilities that GRC picks a particular branch of the binary tree and terminates at step $d$, or does not terminate up to and including step $d$, respectively.

**Proposition 3** (Acceptance and rejection probabilities). *Let $V_d$ be the event that GRC does not terminate up to and including step $d$ and $W_d$ be the event that it terminates at step $d$. Let $S_{0:d} = B_{0:d}$ denote the event that the sequence of the first $d$ bounds produced is $B_{0:d}$. Then*

$$\mathbb{P}(V_d, S_{0:d} = B_{0:d}) = 1 - T_d(\mathcal{X}, B_{0:d}), \qquad \text{for } d = 0, 1, \ldots, \tag{30}$$
$$\mathbb{P}(W_{d+1}, S_{0:d} = B_{0:d}) = A_{d+1}(\mathcal{X}, B_{0:d}), \qquad \text{for } d = 0, 1, \ldots. \tag{31}$$

*Proof.* First we consider the probability that GRC terminates at step $k + 1$ given that it has not terminated up to and including step $d$, that is the quantity $\mathbb{P}(W_{k+1} \mid V_k, S_{0:k} = B_{0:k})$. By definition 5, this probability is given by integrating the acceptance probability $\beta_{k+1}(x, B_{0:k})$ over $x \in \mathcal{X}$, with respect to the measure $P|_{B_k}/P(B_k)$, that is

$$\mathbb{P}(W_{k+1} \mid V_k, S_{0:k} = B_{0:k}) = \int_{x \in B_k} dP(x) \frac{\beta_{k+1}(x, B_{0:k})}{P(B_k)} \tag{32}$$

$$= \int_{x \in \mathcal{X}} dP(x) \frac{\beta_{k+1}(x, B_{0:k})}{P(B_k)} \tag{33}$$

$$= \int_{x \in \mathcal{X}} dP(x) \frac{\alpha_{k+1}(x, B_{0:k})}{1 - T_k(\mathcal{X}, B_{0:k})} \tag{34}$$

$$= \frac{A_{k+1}(\mathcal{X}, B_{0:k})}{1 - T_k(\mathcal{X}, B_{0:k})}, \tag{35}$$

Now, we show the result by induction on $d$, starting from the base case of $d = 0$. **Base case:** For $d = 0$, by the definition of GRC (definition 5) $S_0 = Z_{I_0} = \mathcal{X}$, so

$$\mathbb{P}(V_0, S_0 = B_0) = 1 \text{ and } T_0(\mathcal{X}, B_0) = 0, \tag{36}$$

which show the base case for eq. (30). Now, plugging in $k = 0$ in eq. (35) we obtain

$$\mathbb{P}(W_1, S_0 = B_0) = \mathbb{P}(W_1 \mid V_0, S_0 = B_0) = \frac{A_1(\mathcal{X}, B_0)}{1 - T_0(\mathcal{X}, B_0)} = A_1(\mathcal{X}, B_0) \tag{37}$$

where we have used the fact that $T_0(\mathcal{X}, B_0) = 0$, showing the base case for eq. (31).

**Inductive step:** Suppose that for all $k = 0, 1, 2, \ldots, d$ it holds that

$$\mathbb{P}(V_d, S_{0:k} = B_{0:k}) = 1 - T_d(\mathcal{X}, B_{0:k}) \text{ and } \mathbb{P}(W_{k+1}, S_{0:k} = B_{0:k}) = A_{k+1}(\mathcal{X}, B_{0:k}). \tag{38}$$

Setting $k = d$ in eq. (35), we obtain

$$\mathbb{P}(W'_{d+1} \mid V_d, S_{0:d} = B_{0:d}) = \frac{1 - T_d(\mathcal{X}, B_{0:d}) - A_{d+1}(\mathcal{X}, B_{0:d})}{1 - T_d(\mathcal{X}.B_{0:d})}, \tag{39}$$

and using the inductive hypothesis from eq. (38), we have

$$\mathbb{P}(V_{d+1}, S_{0:d} = B_{0:d}) = \mathbb{P}(W'_{d+1}, V_d, S_{0:d} = B_{0:d}) = 1 - T_d(\mathcal{X}, B_{0:d}) - A_{d+1}(\mathcal{X}, B_{0:d}). \quad (40)$$

Now, $B_d = Z_n$ for some $n \in \mathbb{N}^+$. Denote $B_d^L := Z_{2n}$ and $B_d^R := Z_{2n+1}$. Then, by the product rule

$$\mathbb{P}(V_{d+1}, S_{0:d} = B_{0:d}, S_{d+1} = B_d^R) = \quad (41)$$

$$= \mathbb{P}(S_{d+1} = B_d^R \mid V_{d+1}, S_{0:d} = B_{0:d})\mathbb{P}(V_{d+1}, S_{0:d} = B_{0:d}) \quad (42)$$

$$= \frac{Q(B_d^R) - T_d(B_d^R, B_{0:d}) - A_{d+1}(B_d^R, B_{0:d})}{Q(B_d) - T_d(B_d, B_{0:d}) - A_{d+1}(B_d, B_{0:d})}\mathbb{P}(V_{d+1}, S_{0:d} = B_{0:d}) \quad (43)$$

$$= \frac{Q(B_d^R) - T_d(B_d^R, B_{0:d}) - A_{d+1}(B_d^R, B_{0:d})}{\underbrace{Q(\mathcal{X})}_{=1} - T_d(\mathcal{X}, B_{0:d}) - A_{d+1}(\mathcal{X}, B_{0:d})}\mathbb{P}(V_{d+1}, B_{0:d} = B_{0:d}) \quad (44)$$

$$= Q(B_d^R) - T_d(B_d^R, B_{0:d}) - A_{d+1}(B_d^R, B_{0:d}) \quad (45)$$

$$= 1 - T_{d+1}(\mathcal{X}, B_{0:d+1}) \quad (46)$$

where we have written $B_{0:d+1} = (B_0, \ldots, B_d, B_d^R)$. Above, to go from 41 to 42 we used the definition of conditional probability, to go from 42 to 43 we used the definition in 19, to go from 43 to 44 we used the fact that for $k = 0, 1, 2, \ldots$, it holds that

$$Q(\mathcal{X}) - T_k(\mathcal{X}, B_{0:k}) - A_{k+1}(\mathcal{X}, B_{0:k}) = Q(B_k) - T_k(B_k, B_{0:k}) - A_{k+1}(B_k, B_{0:k}) +$$
$$+ Q(B'_k) - \underbrace{T_k(B'_k, B_{0:k})}_{= Q(B'_k)} - \underbrace{A_{k+1}(B'_k, B_{0:k})}_{= 0} \quad (47)$$

$$= Q(B_k) - T_d(B_k, B_{0:k}) - A_{k+1}(B_k, B_{0:k}), \quad (48)$$

from 44 to 45 we have used eq. (40), and lastly from 45 to 46 we have again used eq. (48). Equation (46) similarly holds if $B_{d+1} = B_d^R$ by $B_{d+1} = B_d^L$, so we arrive at

$$\mathbb{P}(V_{d+1}, B_{0:d+1} = B_{0:d+1}) = 1 - T_{d+1}(\mathcal{X}, B_{0:d+1}), \quad (49)$$

which shows the inductive step for eq. (30). Further, we have

$$\mathbb{P}(W_{d+2}, B_{0:d+1} = B_{0:d+1}) = \mathbb{P}(W_{d+2} \mid V_{d+1}, B_{0:d+1} = B_{0:d+1})\mathbb{P}(V_{d+1}, B_{0:d+1} = B_{0:d+1}) \quad (50)$$

and also by setting $k = d + 1$ in eq. (35) we have

$$\mathbb{P}(W_{d+2} \mid V_{d+1}, B_{0:d+1} = B_{0:d+1}) = \frac{A_{d+2}(\mathcal{X}, B_{0:d+1})}{1 - T_{d+1}(\mathcal{X}, B_{0:d+1})}. \quad (51)$$

Combining eq. (49) and eq. (51) we arrive at

$$\mathbb{P}(W_{d+2}, B_{0:d+1} = B_{0:d+1}) = A_{d+2}(\mathcal{X}, B_{0:d+1}), \quad (52)$$

which is the inductive step for eq. (31). Putting eqs. (49) and (52) together shows the result. □

We now turn to defining and deriving the form of the measure $\tau_D$. We will define $\tau_D$ to be the measure such that for any $S \in \Sigma$, the probability that GRC terminates up to and including step $D$ and returns a sample within $S$ is given by $\tau_D(S)$. We will also show that $\tau_D$ is non-increasing in $D$.

**Lemma 1** (Density of samples generated by GRC). *The probability that GRC terminates by step $D \geq 1$ and produces a sample in $S$ is given by the measure*

$$\tau_D(S) = \sum_{d=1}^{D} \sum_{k=1}^{2^{d-1}} A_d(S, \hat{Z}_{d,k}), \quad (53)$$

*where $\hat{Z}_{D,k}$ is the ancestor branch of $Z_{D,k}$ as defined in eq. (29). Further, $\tau_D$ is non-decreasing in $D$, that is if $n \leq m$, then $\tau_n(S) \leq \tau_m(S)$ for all $S \in \Sigma$.*

*Proof.* Let $V_d$ be the event that GRC does not terminate up to and including step $d$ and let $W_d(S)$ be the event that GRC terminates at step $d$ and returns a sample in $S$. Then

$$\tau_D(S) = \sum_{d=1}^{D} \mathbb{P}(W_d(S)) \tag{54}$$

$$= \sum_{d=1}^{D} \mathbb{P}(W_d(S), V_{d-1}) \tag{55}$$

$$= \sum_{d=1}^{D} \sum_{k=1}^{2^{d-1}} \mathbb{P}(W_d(S), V_{d-1}, S_{0:d-1} = \hat{Z}_{d,k}) \tag{56}$$

$$= \sum_{d=1}^{D} \sum_{k=1}^{2^{d-1}} \mathbb{P}(W_d(S) \mid V_{d-1}, S_{0:d-1} = \hat{Z}_{d,k}) \, \mathbb{P}(V_{d-1}, S_{0:d-1} = \hat{Z}_{d,k}). \tag{57}$$

Further, the terms in the summand can be expressed as

$$\mathbb{P}(V_{d-1}, S_{0:d-1} = \hat{Z}_{d,k}) = 1 - T_{d-1}(\mathcal{X}, \hat{Z}_{d,k}), \tag{58}$$

$$\mathbb{P}(W_d(S) \mid V_{d-1}, S_{0:d-1} = \hat{Z}_{d,k}) = \int_{x \in S} dP(x) \frac{\beta_d(x, \hat{Z}_{d,k})}{P(Z_{d,k})} \tag{59}$$

$$= \int_{x \in S} dP(x) \frac{\alpha_d(x, \hat{Z}_{d,k})}{1 - T_{d-1}(\mathcal{X}, \hat{Z}_{d,k})} \tag{60}$$

$$= \frac{A_d(S, \hat{Z}_{d,k})}{1 - T_{d-1}(\mathcal{X}, \hat{Z}_{d,k})}, \tag{61}$$

and substituting eqs. (58) and (61) into the sum in eq. (57), we obtain eq. (53). Further, since the inner summand is always non-negative, increasing $D$ adds more non-negative terms to the sum, so $\tau_D$ is also non-decreasing in $D$. $\qquad\square$

Now we turn to proving a few results about the measure $\tau_D$. Lemma 2 shows that $\tau_D \leq Q$ for all $D$. This result implies that $\|Q - \tau_D\|_{TV} = Q(\mathcal{X}) - \tau_D(\mathcal{X})$, which we will use later.

**Lemma 2** ($Q - \tau_D$ is non-negative)**.** *Let $D \in \mathbb{N}^+$. Then $Q - \tau_D$ is a positive measure, that is*

$$Q(S) - \tau_D(S) \geq 0 \text{ for any } S \in \Sigma. \tag{62}$$

*Proof.* Let $S \in \Sigma$ and write

$$Q(S) - \tau_D(S) = \sum_{k=1}^{2^{D-1}} Q(S \cap Z_{D,k}) - \tau_D(S \cap Z_{D,k}) \tag{63}$$

$$= \sum_{k=1}^{2^{D-1}} \left[ Q(S \cap Z_{D,k}) - \sum_{d=1}^{D} \sum_{k'=1}^{2^{D-1}} A_d(S \cap Z_{D,k}, \hat{Z}_{D,k'}) \right] \tag{64}$$

$$= \sum_{k=1}^{2^{D-1}} \left[ Q(S \cap Z_{D,k}) - \sum_{d=1}^{D} A_d(S \cap Z_{D,k}, \hat{Z}_{D,k}) \right] \tag{65}$$

$$= \sum_{k=1}^{2^{D-1}} \left[ Q(S \cap Z_{D,k}) - T_{D-1}(S \cap Z_{D,k}, \hat{Z}_{D,k}) - A_D(S \cap Z_{D,k}, \hat{Z}_{D,k}) \right] \tag{66}$$

We will show that the summand in eq. (66) is non-negative. From the definition in eq. (14) we have

$$\alpha_D(x, \hat{Z}_{D,k}) = \min \left\{ \frac{dQ}{dP}(x) - t_{d-1}(x, \hat{Z}_{D,k}), \frac{1 - T_{D-1}(\mathcal{X}, \hat{Z}_{D,k})}{P(Z_{D,k})} \right\} \tag{67}$$

$$\leq \frac{dQ}{dP}(x) - t_{d-1}(x, \hat{Z}_{D,k}) \tag{68}$$

and integrating both sides of eq. (68) over $S \cap Z_{D,k}$, we obtain

$$A_D(S \cap Z_{D,k}, \hat{Z}_{D,k}) \leq Q(S \cap Z_{D,k}) - T_{D-1}(S \cap Z_{D,k}, \hat{Z}_{D,k}) \tag{69}$$

Putting this together with eq. (66) we arrive at

$$Q(S) - \tau_D(S) \geq 0, \tag{70}$$

which is the required result. $\qquad \square$

Thus far we have derived the form of $\tau_D$, shown that it is non-decreasing in $D$ and that it is no greater than $Q$. As we are interested in the limiting behaviour of $\tau_D$, we next show that its limit, $\tau = \lim_{D \to \infty} \tau_D$, is also a measure. Further, it also holds that $\tau \leq Q$.

**Lemma 3** (Measures $\tau_D$ converge to a measure $\tau \leq Q$). *For each $S \in \Sigma$, $\tau_D(S)$ converges to a limit. Further, the function $\tau : \Sigma \to [0, 1]$ defined as*

$$\tau(S) = \lim_{D \to \infty} \tau_D(S) \tag{71}$$

*is a measure on $(\mathcal{X}, \Sigma)$ and $\tau(S) \leq Q(S)$ for all $S \in \Sigma$.*

*Proof.* First, by lemma 1, $\tau_D(S)$ is non-decreasing in $D$, and bounded above by $Q(S)$ for all $S \in \Sigma$. Therefore, for each $S \in \Sigma$, $\tau_D(S)$ converges to some limit as $D \to \infty$. Define $\tau : \Sigma \to [0, 1]$ as

$$\tau(S) = \lim_{D \to \infty} \tau_D(S), \tag{72}$$

and note that $\tau$ is a non-negative set function for which $\tau(\emptyset) = 0$. By the Vitali-Hahn-Saks theorem (see Corollary 4, p. 160; Dunford & Schwartz, 1988), $\tau$ is also countably additive, so it is a measure. Also, by lemma 2, $\tau_D(S) \leq Q(S)$ for all $D \in \mathbb{N}^+$ and all $S \in \Sigma$, so $\tau(S) \leq Q(S)$ for all $S \in \Sigma$. $\quad \square$

**Definition 9** ($H_{d,k}$, $H_d$ and $H$). *For $d = 1, 2, \ldots$ and $k = 1, \ldots, 2^{d-1}$, we define the sets $H_{d,k}$ as*

$$H_{d,k} = \left\{ x \in Z_{d,k} \;\middle|\; \frac{dQ}{dP}(x) - t_{d-1}(x, \hat{Z}_{d,k}) \geq \frac{1 - T_{d-1}(\mathcal{X}, \hat{Z}_{d,k})}{P(Z_{d,k})} \right\}. \tag{73}$$

*Also, define the sets $H_d$ and $H$ as*

$$H_d = \bigcup_{k=1}^{2^{d-1}} H_{d,k} \quad and \quad H = \bigcap_{d=1}^{\infty} H_d. \tag{74}$$

**Lemma 4** ($T_D(\cdot, \hat{Z}_{D+1,k})$ and $\tau_D$ agree in $Z_{D+1,k}$). *Let $R \in \Sigma$. If $R \subseteq Z_{D+1,k}$, then*

$$\tau_D(R) = T_D(R, \hat{Z}_{D+1,k}). \tag{75}$$

*Proof.* Suppose $R \subseteq Z_{D+1,k}$. First, we have

$$\tau_D(R) = \sum_{d=1}^{D} \sum_{k'=1}^{2^{d-1}} A_d(R, \hat{Z}_{d,k'}) = \sum_{d=1}^{D} A_d(R, (\hat{Z}_{D+1,k})_{1:d}). \tag{76}$$

From the definition of $T_D$ in eq. (22), we have

$$T_D(R, \hat{Z}_{D+1,k}) = T_{D-1}(R \cap Z_{D+1,k}, (\hat{Z}_{D+1,k})_{1:D}) + A_D(R \cap Z_{D+1,k}, (\hat{Z}_{D+1,k})_{1:D}) + \tag{77}$$
$$+ \underbrace{Q(R \cap Z'_{D+1,k})}_{= 0}$$

$$= T_{D-1}(R \cap Z_{D+1,k}, (\hat{Z}_{D+1,k})_{1:D}) + A_D(R \cap Z_{D+1,k}, (\hat{Z}_{D+1,k})_{1:D}) \tag{78}$$

$$= T_{D-1}(R, (\hat{Z}_{D+1,k})_{1:D}) + A_D(R, (\hat{Z}_{D+1,k})_{1:D}) \tag{79}$$

where we have used the assumption that $R \subseteq Z_{D+1,k}$. In a similar manner, applying eq. (79) recursively $D - 1$ more times, we obtain

$$T_D(R, \hat{Z}_{D+1,k}) = \sum_{d=1}^{D} A_d(R, (\hat{Z}_{D+1,k})_{1:d}) = \tau_D(R). \tag{80}$$

which is the required result. $\qquad \square$

**Lemma 5** (Equalities with $Q$, $\tau_D$ and $\tau$). *The following two equalities hold*

$$Q(\mathcal{X} \setminus H_D) = \tau_D(\mathcal{X} \setminus H_D) \text{ and } Q(\mathcal{X} \setminus H) = \tau(\mathcal{X} \setminus H). \tag{81}$$

*Proof.* Let $R = Z_{D+1,k} \setminus H_{D,k}$. Then, by similar reasoning used to prove eq. (77), we have

$$T_D(R, \hat{Z}_{D+1,k}) = T_{D-1}(R, (\hat{Z}_{D+1,k})_{1:D}) + A_D(R, (\hat{Z}_{D+1,k})_{1:D}) \tag{82}$$

Further, we also have

$$A_D(R, \hat{Z}_{D,k}) = \int_R dP(x)\, \alpha_D(x, \hat{Z}_{D,k}) \tag{83}$$

$$= \int_R dP(x)\, \min\left\{ \frac{dQ}{dP}(x) - t_{D-1}(x, \hat{Z}_{D,k}), \frac{1 - T_{D-1}(\mathcal{X}, \hat{Z}_{D,k})}{P(Z_{D,k})} \right\} \tag{84}$$

$$= \int_R dP(x)\, \left( \frac{dQ}{dP}(x) - t_{D-1}(x, \hat{Z}_{D,k}) \right) \tag{85}$$

$$= Q(R) - T_{D-1}(R, \hat{Z}_{D,k}) \tag{86}$$

where from eq. (84) to eq. (85) we have used the definition of $H_{D,k}$. Then, combining eqs. (82) and (86) and using lemma 4, we arrive at

$$Q(Z_{D+1,k} \setminus H_{D,k}) = T_D(Z_{D+1,k} \setminus H_{D,k}, \hat{Z}_{D+1,k}) = \tau_D(Z_{D+1,k} \setminus H_{D,k}). \tag{87}$$

Now, using the equation above, we have that

$$\tau_D(\mathcal{X} \setminus H_D) = \sum_{k=1}^{2^D} \tau_D(Z_{D+1,k} \setminus H_D) = \sum_{k=1}^{2^D} Q(Z_{D+1,k} \setminus H_D) = Q(\mathcal{X} \setminus H_D). \tag{88}$$

Now, using $\tau_D \leq \tau \leq Q$ and $\tau_D(\mathcal{X} \setminus H_D) = Q(\mathcal{X} \setminus H_D)$, we have that $\tau(\mathcal{X} \setminus H_D) = Q(\mathcal{X} \setminus H_D)$, which is the first part of the result we wanted to show. Taking limits, we obtain

$$Q(\mathcal{X} \setminus H) = \lim_{D \to \infty} Q(\mathcal{X} \setminus H_D) = \lim_{D \to \infty} \tau(\mathcal{X} \setminus H_D) = \tau(\mathcal{X} \setminus H), \tag{89}$$

which is the second part of the required result. $\qquad\square$

## B.3 Breaking down the proof of Theorem 1 in five cases

In definition 10 we introduce the quantities $w_d = Q(\mathcal{X}) - \tau_d(\mathcal{X})$ and $p_d = \mathbb{P}(W_d \mid V_{d-1})$. Then we break down the proof of theorem 1 in five cases. First, in lemma 7 we show that if $p_d \not\to 0$, then $w_d \to 0$. Second, in lemma 8 we show that if $P(H_d) \to 0$, then $w_d \to 0$. In lemma 9 we show an intermediate result, used in the other three cases, which we consider in lemmas 10, 11 and 12. Specifically, in these three cases we show that if $p_d \to 0$ and $P(H_d) \not\to 0$, and assumption 1, 2 or 3 hold respectively, we have $w_d \to 0$. Putting these results together shows theorem 1.

**Definition 10** ($p_d$, $w_{d,k}$ and $w_d$). *Define $p_d = \mathbb{P}(W_d \mid V_{d-1})$. Also define $w_{d,k}$ and $w_d$ as*

$$w_{d,k} \stackrel{\text{def}}{=} Q(Z_{d,k}) - \tau_d(Z_{d,k}), \tag{90}$$

$$w_d \stackrel{\text{def}}{=} \sum_{k=1}^{2^{d-1}} w_{d,k}. \tag{91}$$

**Lemma 6** ($w_d$ non-increasing in $d$). *The sequence $w_d$ is non-negative and non-increasing in $d$.*

*Proof.* Since $\tau_d$ is non-decreasing in $d$ (from lemma 5) and

$$w_d = \sum_{k=1}^{2^{d-1}} Q(Z_{d,k}) - \tau_d(Z_{d,k}) = Q(\mathcal{X}) - \tau_d(\mathcal{X}), \tag{92}$$

it follows that $w_d$ is a non-increasing and non-negative sequence. $\qquad\square$

**Lemma 7** (Case 1). *If $p_d \not\to 0$, then $w_d \to 0$.*

*Proof.* Let $p_d = \mathbb{P}(W_d \mid V_{d-1})$ and suppose $p_d \not\to 0$. Then, there exists $\epsilon > 0$ such that $p_d > \epsilon$ occurs infinitely often. Therefore, there exists an increasing sequence of integers $a_d \in \mathbb{N}$ such that $p_{a_d} > \epsilon$ for all $d \in \mathbb{N}$. Then

$$\tau_{a_d}(\mathcal{X}) = \mathbb{P}\left(\bigcup_{d=1}^{a_d} W_d\right) \tag{93}$$

$$= 1 - \mathbb{P}\left(V_{a_d}\right), \tag{94}$$

$$= 1 - \prod_{d=1}^{a_d} \mathbb{P}\left(V_d \mid V_{d-1}\right), \tag{95}$$

$$= 1 - \prod_{d=1}^{a_d} (1 - p_d), \tag{96}$$

$$\geq 1 - (1 - \epsilon)^d \to 1 \text{ as } d \to \infty. \tag{97}$$

Therefore, $\tau_d(\mathcal{X}) \to 1$ as $d \to \infty$, which implies that $||Q - \tau_d||_{TV} \to 0$. $\qquad\square$

**Lemma 8** (Case 2). *If $P(H_d) \to 0$, then $w_d \to 0$.*

*Proof.* Suppose $P(H_d) \to 0$. Since $Q \ll P$, we have $Q(H) = 0$, and since $Q \geq \tau \geq 0$ (by lemma 3), we also have $\tau(H) = 0$. Therefore

$$\lim_{d \to \infty} w_d = \lim_{d \to \infty} ||Q - \tau_d||_{TV} \tag{98}$$

$$= Q(\mathcal{X}) - \tau(\mathcal{X}) \tag{99}$$

$$= \underbrace{Q(\mathcal{X} \setminus H) - \tau(\mathcal{X} \setminus H)}_{= 0 \text{ from lemma } 5} + \underbrace{Q(H)}_{= 0} - \underbrace{\tau(H)}_{= 0} \tag{100}$$

$$= 0 \tag{101}$$

which is the required result. $\qquad\square$

**Lemma 9** (An intermediate result). *If $p_d \to 0$ and $w_d \not\to 0$ as $d \to \infty$, then*

$$\sum_{k=1}^{2^{d-1}} \frac{P(H_{d,k})}{P(Z_{d,k})} w_{d,k} \to 0 \text{ as } d \to \infty. \tag{102}$$

*Proof.* Suppose that $p_d = \mathbb{P}(W_d \mid V_{d-1}) \to 0$ and $w_d \not\to 0$. Then

$$\mathbb{P}(W_d \mid V_{d-1}) \geq \mathbb{P}(W_d(H_d) \mid V_{d-1}) \tag{103}$$

$$= \sum_{k=1}^{2^{d-1}} \mathbb{P}\left(W_d(H_{d,k}) \mid V_{d-1}\right) \tag{104}$$

$$= \sum_{k=1}^{2^{d-1}} \mathbb{P}\left(W_d(H_{d,k}), S_{0:d-1} = \hat{Z}_{d,k} \mid V_{d-1}\right) \tag{105}$$

$$= \sum_{k=1}^{2^{d-1}} \mathbb{P}\left(W_d(H_{d,k}) \mid V_{d-1}, S_{0:d-1} = \hat{Z}_{d,k}\right) \mathbb{P}\left(S_{0:d-1} = \hat{Z}_{d,k} \mid V_{d-1}\right) \tag{106}$$

$$= \sum_{k=1}^{2^{d-1}} \frac{P(H_{d,k})}{P(Z_{d,k})} \mathbb{P}\left(S_{0:d-1} = \hat{Z}_{d,k} \mid V_{d-1}\right) \tag{107}$$

$$= \sum_{k=1}^{2^{d-1}} \frac{P(H_{d,k})}{P(Z_{d,k})} \frac{w_{d,k}}{w_d} \to 0. \tag{108}$$

In addition, if $w_d \not\to 0$, then since $0 \leq w_d \leq 1$ we have

$$\sum_{k=1}^{2^{d-1}} \frac{P(H_{d,k})}{P(Z_{d,k})} w_{d,k} \to 0. \tag{109}$$

which is the required result. $\qquad\square$

**Lemma 10** (Case 3). *Suppose that $p_d \to 0$, $P(H_d) \not\to 0$ and assumption 1 holds. Then $w_d \to 0$.*

*Proof.* Suppose that $p_d \to 0$, $P(H_d) \not\to 0$. Suppose also that assumption 1 holds, meaning there exists $M \in \mathbb{R}$ such that $dQ/dP(x) < M$ for all $x \in \mathcal{X}$. Then for any $S \in \Sigma$, we have

$$\frac{Q(S) - \tau(S)}{P(S)} \leq \frac{Q(S)}{P(S)} = \frac{\int_S \frac{dQ}{dP} dP}{P(S)} \leq M \frac{\int_S dP}{P(S)} = M \implies \frac{Q(S) - \tau(S)}{M} \leq P(S). \tag{110}$$

Further, we have

$$\sum_{k=1}^{2^{d-1}} \frac{P(H_{d,k})}{P(Z_{d,k})} w_{d,k} \geq \sum_{k=1}^{2^{d-1}} \frac{P(H_{d,k})}{P(Z_{d,k})} \left(Q(H_{d,k}) - \tau(H_{d,k})\right) \tag{111}$$

$$\geq \frac{1}{M} \sum_{k=1}^{2^{d-1}} \frac{\left(Q(H_{d,k}) - \tau(H_{d,k})\right)^2}{P(Z_{d,k})} \tag{112}$$

$$\geq \frac{1}{M} \sum_{k=1}^{2^{d-1}} \frac{\left(Q(H \cap H_{d,k}) - \tau(H \cap H_{d,k})\right)^2}{P(Z_{d,k})} \tag{113}$$

$$\geq \frac{1}{M} \sum_{k=1}^{2^{d-1}} \frac{\Delta_{d,k}^2}{P(Z_{d,k})} \tag{114}$$

$$= \frac{1}{M} \Phi_d \tag{115}$$

$$\to 0, \tag{116}$$

where in the second inequality we have used eq. (110) and we have defined

$$\Delta_{d,k} \stackrel{\text{def}}{=} Q(H \cap H_{d,k}) - \tau(H \cap H_{d,k}), \tag{117}$$

$$\Phi_d \stackrel{\text{def}}{=} \sum_{k=1}^{2^{d-1}} \frac{\Delta_{d,k}^2}{P(Z_{d,k})}. \tag{118}$$

Now note that the sets $H \cap H_{d+1,2k}$ and $H \cap H_{d+1,2k+1}$ partition the set $H \cap H_{d,k}$. Therefore

$$\Delta_{d,k} = \Delta_{d+1,2k} + \Delta_{d+1,2k+1}. \tag{119}$$

By the definition of $\Phi_d$ in eq. (118), we can write

$$\Phi_{d+1} = \sum_{k=1}^{2^d} \frac{\Delta_{d,k}^2}{P(Z_{d+1,k})} = \sum_{k=1}^{2^{d-1}} \left[ \frac{\Delta_{d+1,2k}^2}{P(Z_{d+1,2k})} + \frac{\Delta_{d+1,2k+1}^2}{P(Z_{d+1,2k+1})} \right], \tag{120}$$

where we have written the sum over $2^d$ terms as a sum over $2^{d-1}$ pairs of terms. We can rewrite the summand on the right hand side as

$$\frac{\Delta_{d+1,2k}^2}{P(Z_{d+1,2k})} + \frac{\Delta_{d+1,2k+1}^2}{P(Z_{d+1,2k+1})} = \frac{\Delta_{d+1,2k}^2}{P(Z_{d+1,2k})} + \frac{(\Delta_{d,k} - \Delta_{d+1,2k})^2}{P(Z_{d+1,2k+1})} \tag{121}$$

$$= \Delta_{d,k}^2 \left[ \frac{\rho^2}{P(Z_{d+1,2k-1})} + \frac{(1-\rho)^2}{P(Z_{d+1,2k})} \right] \tag{122}$$

$$= \Delta_{d,k}^2 \, g(\rho) \tag{123}$$

where in eq. (121) we have used eq. (119), from eq. (121) to eq. (122) we defined the quantity $\rho = \Delta_{d+1,2k}/\Delta_{d,k}$, and from eq. (122) to eq. (123) we have defined $g : [0,1] \to \mathbb{R}$ as

$$g(r) \stackrel{\text{def}}{=} \frac{r^2}{P(Z_{d+1,2k})} + \frac{(1-r)^2}{P(Z_{d+1,2k+1})}. \tag{124}$$

The first and second derivatives of $g$ are

$$\frac{dg}{dr} = \frac{2r}{P(Z_{d+1,2k})} - \frac{2(1-r)}{P(Z_{d+1,2k+1})}, \tag{125}$$

$$\frac{d^2g}{dr^2} = \frac{2}{P(Z_{d+1,2k})} + \frac{2}{P(Z_{d+1,2k+1})} > 0, \tag{126}$$

so $g$ has a single stationary point that is a minimum, at $r = r_{\min}$, which is given by

$$r_{\min} := \frac{P(Z_{d+1,2k})}{P(Z_{d+1,2k}) + P(Z_{d+1,2k+1})}. \tag{127}$$

Plugging this back in $g$, we obtain

$$g(r_{\min}) = \frac{1}{P(Z_{d+1,2k}) + P(Z_{d+1,2k+1})} = \frac{1}{P(Z_{d,k})}, \tag{128}$$

which implies that

$$\frac{\Delta_{d+1,2k}^2}{P(Z_{d+1,2k})} + \frac{\Delta_{d+1,2k+1}^2}{P(Z_{d+1,2k+1})} \geq \frac{\Delta_{d,k}^2}{P(Z_{d,k})}. \tag{129}$$

Therefore

$$\Phi_{d+1} = \sum_{k=1}^{2^d} \frac{\Delta_{d,k}^2}{P(Z_{d+1,k})} \geq \sum_{k=1}^{2^{d-1}} \frac{\Delta_{d,k}^2}{P(Z_{d,k})} = \Phi_d, \tag{130}$$

but since $\Phi_d \to 0$, this is only possible if $\Phi_d = 0$ for all $d$, including $d = 1$, which would imply that

$$\Delta_{1,1} = Q(H \cap H_{1,1}) - \tau(H \cap H_{1,1}) = Q(H) - \tau(H) = 0, \tag{131}$$

which, together with lemma 5, implies that

$$Q(\mathcal{X}) - \tau(\mathcal{X}) = Q(H) - \tau(H) = 0, \tag{132}$$

and therefore $w_d = ||Q - \tau_d||_{TV} \to 0$. $\qquad \square$

**Lemma 11** (Case 4). *Suppose that $p_d \to 0$, $P(H_d) \not\to 0$ and assumption 3 holds. Then $w_d \to 0$.*

*Proof.* Suppose that $p_d \to 0$, $P(H_d) \not\to 0$. Suppose also that assumption that assumption 3 holds, meaning that for each $d$, we have $w_{d,k} > 0$ for exactly one value of $k = k_d$, and $w_{d,k} = 0$ for all other $k \neq k_d$. In this case, it holds that $H_{d,k} = \emptyset$ for all $k \neq k_d$ and $H_d = H_{d,k_d}$. Since $P(H_d) \not\to 0$ and $P(H_d)$ is a decreasing sequence, it converges to some positive constant. We also have

$$p_d \geq \sum_{k=1}^{2^{d-1}} \frac{P(H_{d,k})}{P(Z_{d,k})} w_{d,k} = \frac{P(H_{d,k_d})}{P(Z_{d,k_d})} w_{d,k_d} = \frac{P(H_{d,k_d})}{P(Z_{d,k_d})} w_d \geq P(H_d)\, w_d \to 0, \quad (133)$$

which can only hold if $w_d \to 0$, arriving at the result. $\qquad\square$

**Lemma 12** (Case 5). *Suppose that $p_d \to 0$, $P(H_d) \not\to 0$ and assumption 3 holds. Then $w_d \to 0$.*

*Proof.* Suppose that $p_d \to 0$, $P(H_d) \not\to 0$ and assumption 3 holds. Since each $x \in \mathcal{X}$ belongs to exactly one $Z_{d,k}$ we can define the function $B_d : \mathcal{X} \to \Sigma$ as

$$B_d(x) = Z_{d,k} \text{ such that } x \in Z_{d,k}. \quad (134)$$

Using this function we can write

$$p_d \geq \sum_{k=1}^{2^{d-1}} \frac{P(H_{d,k})}{P(Z_{d,k})} w_{d,k} = \sum_{k=1}^{2^{d-1}} P(H_{d,k}) \frac{Q(Z_{d,k}) - \tau_d(Z_{d,k})}{P(Z_{d,k})} = \int_{H_d} dP \, \frac{Q(B_d(x)) - \tau_d(B_d(x))}{P(B_d(x))}.$$

Now, because the sets $H_d$ are measurable, their intersection $H := \cap_{d=1}^{\infty} H_d$ is also measurable. We can therefore lower bound the integral above as follows

$$\int_{H_d} dP \, \frac{Q(B_d(x)) - \tau_d(B_d(x))}{P(B_d(x))} \geq \int_H dP \, \frac{Q(B_d(x)) - \tau_d(B_d(x))}{P(B_d(x))} \quad (135)$$

$$\geq \int_H dP \, \frac{Q(B_d(x)) - \tau(B_d(x))}{P(B_d(x))}, \quad (136)$$

where the first inequality holds as the integrand is non-negative and we are constraining the integration domain to $H \subseteq H_d$, and the second inequality holds because $\tau_d(S) \leq \tau(S)$ for any $S \in \Sigma$. Define $\mathcal{C}$ to be the set of all intersections of nested partitions, with non-zero mass under $P$

$$\mathcal{C} = \left\{ \bigcap_{d=0}^{\infty} Z_{d,k_d} : P\left( \bigcap_{d=0}^{\infty} Z_{d,k_d} \right) > 0, k_0 = 1, k_{d+1} = 2k_d \text{ or } k_{d+1} = 2k_d + 1 \right\}, \quad (137)$$

and note that all of its elements are pairwise disjoint. Each of the elements of $\mathcal{C}$ is a measurable set because it is a countable intersection of measurable sets. In addition, $\mathcal{C}$ is a countable set, which can be shown as follows. Define the sets $\mathcal{C}_n$ as

$$\mathcal{C}_n = \left\{ E \in \mathcal{C} : 2^{-n-1} < P(E) \leq 2^{-n} \right\} \text{ for } n = 0, 1, \dots \quad (138)$$

and note that their union equals $\mathcal{C}$. Further, note that each $\mathcal{C}_n$ must contain a finite number of elements. That is because if $\mathcal{C}_n$ contained an infinite number of elements, say $E_1, E_2, \dots \in \mathcal{C}_n$, then

$$P(\mathcal{X}) \geq P\left( \bigcup_{k=1}^{\infty} E_k \right) = \sum_{k=1}^{\infty} P(E_k) > \sum_{k=1}^{\infty} 2^{-n-1} \to \infty, \quad (139)$$

where the first equality holds because $P$ is an additive measure and the $E_n$ terms are disjoint, and the second inequality follows because $E_k \in \mathcal{C}_n$ so $P(E_k) > 2^{-n-1}$. This results in a contradiction because $P(\mathcal{X}) = 1$, so each $\mathcal{C}_n$ must contain a finite number of terms. Therefore, $\mathcal{C}$ is a countable union of finite sets, which is also countable. This implies that the union of the elements of $\mathcal{C}$, namely $C = \cup_{C' \in \mathcal{C}} C'$ is a countable union of measurable sets and therefore also measurable. Since $C$ is measurable, $H \setminus C$ is also measurable and we can rewrite the integral in eq. (135) as

$$p_d \geq \int_H dP \, \frac{Q(B_d(x)) - \tau(B_d(x))}{P(B_d(x))} \quad (140)$$

$$= \int_{H \cap C} dP \, \frac{Q(B_d(x)) - \tau(B_d(x))}{P(B_d(x))} + \int_{H \setminus C} dP \, \frac{Q(B_d(x)) - \tau(B_d(x))}{P(B_d(x))} \quad (141)$$

$$\to 0 \quad (142)$$

Since both terms above are non-negative and their sum converges to 0, the terms must also individually converge to 0. Therefore, for the first term, we can write

$$\lim_{d\to\infty} \int_{H\cap C} dP\, \frac{Q(B_d(x)) - \tau(B_d(x))}{P(B_d(x))} = \liminf_{d\to\infty} \int_{H\cap C} dP\, \frac{Q(B_d(x)) - \tau(B_d(x))}{P(B_d(x))} = 0. \quad (143)$$

Similarly to $B_d$ defined in eq. (134), let us define $B : C \to \Sigma$ as

$$B(x) = C' \in \mathcal{C} \text{ such that } x \in C'. \quad (144)$$

Applying Fatou's lemma (4.3.3, p. 131; Dudley, 2018) to eq. (143), we obtain

$$\liminf_{d\to\infty} \int_{H\cap C} dP\, \frac{Q(B_d(x)) - \tau(B_d(x))}{P(B_d(x))} \geq \int_{H\cap C} dP\, \liminf_{d\to\infty} \frac{Q(B_d(x)) - \tau(B_d(x))}{P(B_d(x))} \quad (145)$$

$$= \int_{H\cap C} dP\, \frac{Q(B(x)) - \tau(B(x))}{P(B(x))} \quad (146)$$

$$= 0, \quad (147)$$

where from eq. (145) to eq. (146) we have used the fact that $P(B_d(x)) > 0$ whenever $x \in C$ and also that $B_1(x) \supseteq B_2(x) \supseteq \dots$. Now we can re-write this integral as a sum, as follows. Let the elements of $\mathcal{C}$, which we earlier showed is countable, be $C_1, C_2, \dots$ and write

$$\int_{H\cap C} dP\, \frac{Q(B(x)) - \tau(B(x))}{P(B(x))} = \sum_{n=1}^{\infty} \int_{H\cap C_n} dP\, \frac{Q(B(x)) - \tau(B(x))}{P(B(x))} \quad (148)$$

$$= \sum_{n=1}^{\infty} \frac{P(H\cap C_n)}{P(C_n)} \left( Q(C_n) - \tau(C_n) \right) \quad (149)$$

$$= 0. \quad (150)$$

Now, from lemma 5, we have

$$\sum_{n=1}^{\infty} \frac{P(H\cap C_n)}{P(C_n)} \left( Q(C_n) - \tau(C_n) \right) = \sum_{n=1}^{\infty} \frac{P(H\cap C_n)}{P(C_n)} \left( Q(H\cap C_n) - \tau(H\cap C_n) \right) = 0, \quad (151)$$

which in turn implies that for each $n = 1, 2, \dots$, we have either $Q(H\cap C_n) - \tau(H\cap C_n) = 0$ or $P(H\cap C_n) = 0$. However, the latter case also implies $Q(H\cap C_n) - \tau(H\cap C_n) = 0$ because $Q \ll P$, so $Q(H\cap C_n) - \tau(H\cap C_n) = 0$ holds for all $n$. Therefore

$$\tau(H\cap C) = \sum_{n=1}^{\infty} \tau(H\cap C_n) = \sum_{n=1}^{\infty} Q(H\cap C_n) = Q(H\cap C). \quad (152)$$

Returning to the second term in the right hand of eq. (141), and again applying Fatou's lemma

$$\liminf_{d\to\infty} \int_{H\setminus C} dP\, \frac{Q(B_d(x)) - \tau(B_d(x))}{P(B_d(x))} \geq \int_{H\setminus C} dP\, \liminf_{d\to\infty} \frac{Q(B_d(x)) - \tau(B_d(x))}{P(B_d(x))}. \quad (153)$$

Now, since $Z$ has the nice-shrinking property from assumption 3, we can apply a standard result from measure theory and integration Rudin (1986, given in Theorem 7.10, p. 140), to show that the following limit exists and the following equalities are satisfied

$$\lim_{d\to\infty} \frac{Q(B_d(x)) - \tau(B_d(x))}{P(B_d(x))} = \lim_{d\to\infty} \frac{1}{P(B_d(x))} \int_{B_d} dP\, \left( \frac{dQ}{dP}(x) - \frac{d\tau}{dP}(x) \right) \quad (154)$$

$$= \frac{dQ}{dP}(x) - \frac{d\tau}{dP}(x) \quad (155)$$

Inserting 155 into eq. (153), we obtain

$$\liminf_{d\to\infty} \int_{H\setminus C} dP\, \frac{Q(B_d(x)) - \tau(B_d(x))}{P(B_d(x))} \geq \int_{H\setminus C} dP\, \left( \frac{dQ}{dP}(x) - \frac{d\tau}{dP}(x) \right) = 0, \quad (156)$$

which in turn implies that

$$\frac{dQ}{dP}(x) - \frac{d\tau}{dP}(x) = 0 \ \ P\text{-almost-everywhere on } H \setminus C, \quad (157)$$

or equivalently that $Q(H \setminus C) = \tau(H \setminus C)$. Combining this with the fact that $Q(\mathcal{X} \setminus H) = \tau(\mathcal{X} \setminus H)$ and our earlier result that $Q(H \cap C) = \tau(H \cap C)$, we have

$$||Q - \tau||_{TV} = Q(\mathcal{X} \setminus H) - \tau(\mathcal{X} \setminus H) + Q(H \setminus C) - \tau(H \setminus C) + Q(H \cap C) - \tau(H \cap C) = 0,$$

which is equivalent to $w_d = ||Q - \tau_d||_{TV} \to 0$, that is the required result. $\qquad \square$

**Theorem** (Correcness of GRC). *If any one of the assumptions 1, 2 or 3 holds, then*

$$||Q - \tau_d||_{TV} \to 0 \ \ as \ \ d \to \infty. \tag{158}$$

*Proof.* If $p_d \to 0$, then $w_d \to 0$ by lemma 7. If $P(H_d) \to 0$, then $w_d \to 0$ by lemma 8. Therefore suppose that $p_d \not\to 0$ and $P(H_d) \not\to 0$. Then if any one of assumptions 1, 2 or 3 holds, we can conclude from lemma 10, 11 or 12 respectively, that $||Q - \tau_d||_{TV} \to 0$. $\qquad \square$

# C Optimality of GRCS

---

**Algorithm 3** GRCS with arthmetic coding for the heap index.

---

**Require:** Target $Q$, proposal $P$ over $\mathbb{R}$ with unimodal density ratio $r = dQ/dP$ with mode $\mu$.
1: $d \leftarrow 0, T_0 \leftarrow 0, L_0 \leftarrow 0$
2: $I_0 \leftarrow 1, S_1 \leftarrow \mathbb{R}$
3: **while** True **do**
4:     $X_{I_d} \sim P|_{S_d}/P(S_d)$
5:     $U_{I_d} \sim \text{Uniform}(0,1)$
6:     $\beta_{I_d} \leftarrow \texttt{clip}\left(P(S_d) \cdot \frac{r(X_{I_d}) - L_d}{1 - T_d}, 0, 1\right)$         $\triangleright \texttt{clip}(y,a,b) \stackrel{def}{=} \max\{\min\{y,b\}, a\}$
7:     **if** $U_{I_d} \leq \beta_{d+1}$ **then**
8:         **return** $X_{I_d}, I_d$
9:     **end if**
10:    **if** $X_{I_d} > \mu$ **then**
11:        $I_{d+1} \leftarrow 2I_d$
12:        $S_{d+1} \leftarrow S_d \cap (-\infty, X_{I_d})$
13:    **else**
14:        $I_{d+1} \leftarrow 2I_d + 1$
15:        $S_{d+1} \leftarrow S_d \cap (X_{I_d}, \infty)$
16:    **end if**
17:    $L_{d+1} \leftarrow L_d + T_d/P(S_d)$
18:    $T_{d+1} \leftarrow \mathbb{P}_{Y \sim Q}[r(Y) \geq L_{d+1}] - L_{d+1} \cdot \mathbb{P}_{Y \sim P}[r(Y) \geq L_{d+1}]$
19:    $d \leftarrow d + 1$
20: **end while**

---

In this section, we prove Theorems 2 and 3. We are only interested in continuous distributions over $\mathbb{R}$ with unimodal density ratio $dQ/dP$ for these theorems. Hence, we begin by specializing Algorithm 2 to this setting, shown in Algorithm 3. For simplicity, we also dispense with the abstraction of partitioning processes and show the bound update process directly. Furthermore, we also provide an explicit form for the `AcceptProb` and `RuledOutMass` functions.

Before we move on to proving our proposed theorems, we first prove two useful results. First, we bound the negative log $P$-mass of the bounds with which Algorithm 3 terminates.

**Lemma 13.** *Let $Q$ and $P$ be distributions over $\mathbb{R}$ with unimodal density ratio $r = dQ/dP$, given to Algorithm 3 as the target and proposal distribution as input, respectively. Let $d \geq 0$ and let $X_{1:d} \stackrel{def}{=} X_1, \ldots, X_d$ denote the samples simulated by Algorithm 3 up to step $d+1$, where for $d = 0$ we define the empty list as $X_{1:0} = \emptyset$. Let $S_d$ denote the bounds at step $d + 1$. Then,*

$$-\sum_{j=0}^{d} A_{j+1}(\mathbb{R}, S_{0:d}) \cdot \log P(S_j) \leq D_{\text{KL}}[Q\|P] + \log e. \tag{159}$$

*Proof.* For brevity, we will write $A_d = A_d(\mathbb{R}, S_{0:d})$ and $T_d = T_d(\mathbb{R}, S_{0:d})$. Furthermore, as in Algorithm 3, we define

$$L_d \stackrel{def}{=} \sum_{j=0}^{d-1} \frac{1 - T_j}{P(S_j)} \quad \text{with} \quad L_0 = 0. \tag{160}$$

Note that $X_{1:d}$ is well-defined for all $d \geq 0$ since we could remove the return statement from the algorithm to simulate the bounds it would produce up to an arbitrary step $d$. Now, note that by Proposition 3 we have $\mathbb{P}[D = d \mid X_{1:d}] = A_{d+1}(\mathbb{R}, S_{0:d})$. Now, fix $d \geq 0$ and bounds $S_{0:d}$, and let $x \in \mathbb{R}$ be such that $\alpha_{d+1}(x) > 0$ which holds whenever $r(x) \geq L_d$. From this, for $d \geq 1$ we find

$$r(x) \geq \sum_{j=0}^{d-1} \frac{1 - T_j}{P(S_j)} \tag{161}$$

$$\geq \frac{1 - T_{d-1}}{P(S_{d-1})}, \tag{162}$$

where the second inequality follows from the fact that the $(1 - T_j)/P(S_j)$ terms are all positive. taking logs, we get

$$\log r(x) - \log(1 - T_{d-1}) \geq -\log P(S_{d-1}). \tag{163}$$

Now, we consider the expectation of interest:

$$\sum_{j=0}^{d} -A_{j+1} \cdot \log P(S_j) = -\sum_{j=0}^{d} \int_{\mathbb{R}} \alpha_{j+1}(x) \log P(S_j) \, dx \tag{164}$$

$$\overset{\text{eq. }(163)}{\leq} \sum_{j=0}^{d} \int_{\mathbb{R}} \alpha_{j+1}(x)(\log(r(x)) - \log(1 - T_j)) \, dx \tag{165}$$

$$\overset{(a)}{\leq} \int_{\mathbb{R}} \sum_{j=0}^{\infty} \alpha_{j+1}(x) \log r(x) \, dx + \sum_{j=0}^{\infty} A_{j+1} \log \frac{1}{1 - T_j} \tag{166}$$

$$\overset{(b)}{=} \int_{\mathbb{R}} q(x) \log r(x) \, dx + \sum_{j=0}^{\infty} (T_{j+1} - T_j) \log \frac{1}{1 - T_j} \tag{167}$$

$$= D_{\mathrm{KL}}[Q\|P] + \sum_{j=0}^{\infty} (T_{j+1} - T_j) \log \frac{1}{1 - T_j} \tag{168}$$

$$\overset{(c)}{\leq} D_{\mathrm{KL}}[Q\|P] \cdot \log 2 + \int_0^1 \log \frac{1}{1 - t} \, dt \tag{169}$$

$$= D_{\mathrm{KL}}[Q\|P] + \log e. \tag{170}$$

Inequality (a) holds because all terms are positive. This is guaranteed by the fact that for $d \geq 1$, we have $L_d \geq 1$, hence $0 \leq \log L_d \leq r(x)$ whenever Equation (163) holds. Equality (b) follows by the correctness of GRC (Theorem 1), which implies that for all $x \in \mathbb{R}$ we have $\sum_{j=0}^{\infty} \alpha_d(x) = q(x)$, and inequality (c) follows from the facts that $0 \leq T_d \leq 1$ for all $d$ and that the summand in the second term forms a lower-Riemann sum approximation to $-\log(1 - t)$. $\qquad\square$

Second, we consider the contraction rate of the bounds $S_{0:d}$, considered by Algorithm 3.

**Lemma 14.** *Let $Q$ and $P$ be distributions over $\mathbb{R}$ with unimodal density ratio $r = dQ/dP$, given to Algorithm 3 as the target and proposal distribution as input, respectively. Assume $P$ has CDF $F_P$ and the mode of $r$ is at $\mu$. Fix $d \geq 0$ and let $X_{1:d}$ be the samples considered by Algorithm 3 and $S_d$ the bounds at step $d + 1$. Then,*

$$\mathbb{E}_{X_{1:d}}[P(S_d)] \leq \left(\frac{3}{4}\right)^d \tag{171}$$

*Proof.* We prove the claim by induction. For $d = 0$ the claim holds trivially, since $S_0 = \mathbb{R}$, hence $P(S_0) = 1$. Assume now that the claim holds for $d = k - 1$, and we prove the statement for $d = k$. By the law of iterated expectations, we have

$$\mathbb{E}_{X_{1:k}}[P(S_k)] = \mathbb{E}_{X_{1:k-1}}[\mathbb{E}_{X_k | X_{1:k-1}}[P(S_k)]]. \tag{172}$$

Let us now examine the inner expectation. First, assume that $S_{k-1} = (a, b)$ for some real numbers $a < b$ and define $A = F_P(a), B = F_P(B), M = F_P(\mu)$ and $U = F_P(X_k)$. Since $X_k \mid X_{1:k-1} \sim P|_{S_{k-1}}$, by the probability integral transform we have $U \sim \mathrm{Unif}(A, B)$, where $\mathrm{Unif}(A, B)$ denotes the uniform distribution on the interval $(A, B)$. The two possible intervals from which Algorithm 3 will choose are $(a, X_k)$ and $(X_k, b)$, whose measures are $P((a, X_k)) = F_P(X_k) - F_P(a) = U - A$ and similarly $P((X_k, b)) = B - U$. Then, $P(S_k) \leq \max\{U - A, B - U\}$, from which we obtain the bound

$$\mathbb{E}_{X_k | X_{1:k-1}}[P(S_k)] \leq \mathbb{E}_U[\max\{U - A, B - U\}] = \frac{3}{4}(B - A) = \frac{3}{4} P(S_{k-1}). \tag{173}$$

Plugging this into Equation (172), we get

$$\mathbb{E}_{X_{1:k}}[P(S_k)] \leq \frac{3}{4}\mathbb{E}_{X_{1:k-1}}[P(S_{k-1})] \tag{174}$$

$$\leq \frac{3}{4} \cdot \left(\frac{3}{4}\right)^{k-1}, \tag{175}$$

where the second inequality follows from the inductive hypothesis, which finishes the proof. $\square$

**The proof of Theorem 3:** We prove our bound on the runtime of Algorithm 3 first, as this will be necessary for the proof of the bound on the codelength. First, let $D$ be the number of steps Algorithm 3 takes before it terminates minus 1. Then, we will show that

$$\mathbb{E}[D] \leq \frac{1}{\log(4/3)}D_{\mathrm{KL}}[Q\|P] + 4 \tag{176}$$

We tackle this directly. Hence, let

$$\mathbb{E}_D[D] = \lim_{d \to \infty} \mathbb{E}_{X_{1:j}}\left[\sum_{j=1}^{d} j \cdot A_{j+1}\right] \tag{177}$$

$$= \lim_{d \to \infty} \mathbb{E}_{X_{1:j}}\left[\sum_{j=1}^{d} \frac{-j}{\log P(S_j)} \cdot -A_{j+1}\log P(S_j)\right] \tag{178}$$

$$\leq \lim_{d \to \infty} \mathbb{E}_{X_{1:j}}\left[\max_{j \in [1:d]}\left\{\frac{-j}{\log P(S_j)}\right\} \cdot \sum_{j=1}^{d} -A_{j+1}\log P(S_j)\right] \tag{179}$$

$$\overset{\text{lemma 13}}{\leq} (D_{\mathrm{KL}}[Q\|P] + \log e) \cdot \lim_{d \to \infty} \mathbb{E}_{X_{1:j}}\left[\max_{j \in [1:d]}\left\{\frac{-j}{\log P(S_j)}\right\}\right]. \tag{180}$$

To finish the proof, we will now bound the term involving the limit. To do this, note, that for any finite collection of reals $F$, we have $\max_{x \in F}\{x\} = -\min_{x \in F}\{-x\}$, and that for a finite collection of real-valued random variables $\hat{F}$ we have $\mathbb{E}[\min_{\mathbf{x} \in \hat{F}}\{\mathbf{x}\}] \leq \min_{\mathbf{x} \in \hat{F}}\{\mathbb{E}[\mathbf{x}]\}$. Now, we have

$$\lim_{d \to \infty} \mathbb{E}_{X_{1:j}}\left[\max_{j \in [1:d]}\left\{\frac{-j}{\log P(S_j)}\right\}\right] = \lim_{d \to \infty} -\mathbb{E}_{X_{1:j}}\left[\min_{j \in [1:d]}\left\{\frac{j}{\log P(S_j)}\right\}\right] \tag{181}$$

$$\leq \lim_{d \to \infty}\left(-\min_{j \in [1:d]}\left\{\mathbb{E}_{X_{1:j}}\left[\frac{j}{\log P(S_j)}\right]\right\}\right) \tag{182}$$

$$\overset{(a)}{\leq} \lim_{d \to \infty}\left(-\min_{j \in [1:d]}\left\{\frac{j}{\log \mathbb{E}_{X_{1:j}}[P(S_j)]}\right\}\right) \tag{183}$$

$$\overset{\text{lemma 14}}{\leq} \lim_{d \to \infty}\left(-\min_{j \in [1:d]}\left\{\frac{-j}{j\log(4/3)}\right\}\right) \tag{184}$$

$$= \lim_{d \to \infty}\left(\max_{j \in [1:d]}\left\{\frac{1}{\log(4/3)}\right\}\right) \tag{185}$$

$$= \frac{1}{\log(4/3)} \tag{186}$$

Inequality (a) follows from Jensen's inequality. Finally, plugging this back into the previous equation, we get

$$\mathbb{E}[D] \leq \frac{D_{\mathrm{KL}}[Q\|P] + \log e}{\log 4/3} \leq \frac{D_{\mathrm{KL}}[Q\|P]}{\log 4/3} + 4 \tag{187}$$

**Proof of Theorem 2:** For the codelength result, we need to encode the length of the search path and the search path itself. More formally, since the returned sample $X$ is a function of the partition process $Z$, the search path length $D$ and search path $S_{0:D}$, we have

$$\mathbb{H}[X \mid Z] \leq \mathbb{H}[D, S_{0:D}] = \mathbb{H}[D] + \mathbb{H}[S_{0:D} \mid D]. \tag{188}$$

we can encode $D$ using Elias $\gamma$-coding, from which we get

$$\mathbb{H}[D] \leq \mathbb{E}_D[2\log(D+1)] + 1 \tag{189}$$
$$\leq 2\log(\mathbb{E}[D] + 1) + 1 \tag{190}$$
$$\leq 2\log\left(\frac{D_{\mathrm{KL}}[Q\|P] + \log e}{\log(4/3)} + 1\right) + 1 \tag{191}$$
$$\leq 2\log\left(D_{\mathrm{KL}}[Q\|P] + \log e + \log(4/3)\right) + 1 - 2\log\left(\log(4/3)\right) \tag{192}$$
$$\leq 2\log\left(D_{\mathrm{KL}}[Q\|P] + 1\right) + 1 - 2\log\left(\log(4/3)\right) + 2\log(\log e + \log(4/3)) \tag{193}$$
$$\leq 2\log\left(D_{\mathrm{KL}}[Q\|P] + 1\right) + 6. \tag{194}$$

Given the search path length $D$, we can use arithmetic coding (AC) to encode the sequence of bounds $S_{0:D}$ using $-\log P(S_D) + 2$ bits (assuming infinite precision AC). Hence, we have that the average coding cost is upper bounded by

$$\mathbb{H}[S_{0:D} \mid D] \leq \mathbb{E}_D[-\log P(S_D)] + 2 \overset{\text{lemma 13}}{\leq} D_{\mathrm{KL}}[Q\|P] + 5. \tag{195}$$

Putting everything together, we find

$$\mathbb{H}[D, S_{0:D}] \leq D_{\mathrm{KL}}[Q\|P] + 2\log(D_{\mathrm{KL}}[Q\|P] + 1) + 11, \tag{196}$$

as required.

## D  Additional experiments with depth-limited GRC

In this section we show the results of some experiments comparing the approximation bias of depth limited GRCD, to that of depth limited AD$^*$, following the setup of Flamich et al. (2022). Limiting the depth of each algorithm introduces bias in the resulting samples, as these are not guaranteed to be distributed from the target distribution $Q$, but rather from a different distribution $\hat{Q}$. Figure 5 quantifies the effect of limiting the depth on the bias of the resulting samples.

In our experiment we take $Q$ and $P$ to be Gaussian and we fix $D_{\mathrm{KL}}[Q\|P] = 3$ (bits), and consider three different settings of $D_\infty[Q\|P] = 5, 7$ or $9$ (bits), corresponding to each of the panes in fig. 5. For each such setting, we set the depth limit of each of the two algorithms to $D_{\max} = D_{\mathrm{KL}}[Q\|P] + d$ bits, and refer to $d$ as the *number of additional bits*. We then vary the number of additional bits allowed for each algorithm, and estimate the bias of the resulting samples by evaluating the KL divergence between the empirical and the exact target distribution, that is $D_{\mathrm{KL}}[\hat{Q}\|Q]$. To estimate this bias, we follow the method of Pérez-Cruz (2008). For each datapoint shown we draw 200 samples $X \sim \hat{Q}$ and use these to estimate $D_{\mathrm{KL}}[\hat{Q}\|Q]$. We then repeat this for 10 different random seeds, reporting the mean bias and standard error in the bias, across these 10 seeds.

Generally we find that the bias of GRCD is higher than that of AD$^*$. This is likely because AD$^*$ is implicitly performing importance sampling over a set of $2^{D_{\max}+d} - 1$ samples, and returning the one with the highest importance weight. By contrast, GRCD is running rejection sampling up to a maximum of $D_{\max} + d$ steps, returning its last sample if it has not terminated by its $(D_{\max} + d)^{\text{th}}$ step. While it might be possible to improve the bias of depth limited GRCD by considering an alternative way of choosing which sample to return, using for example an importance weighting criterion, we do not examine this here and leave this possibility for future work.

Bias for Gaussian $Q$ and $P$

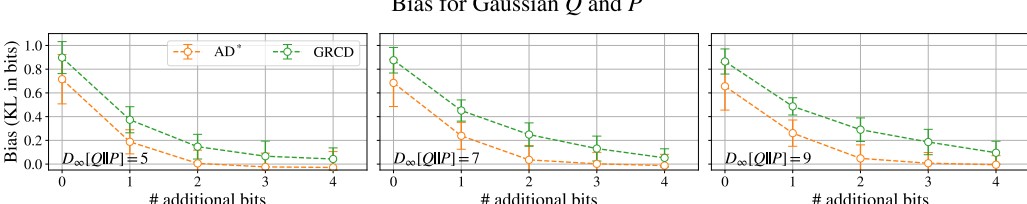

Figure 5: Bias of depth-limited AD$^*$ and GRCD, as a function of the number of additional bit budget given to each algorithm. See text above for discussion.