# OpenReview forum: "Faster Relative Entropy Coding with Greedy Rejection Coding"
_NeurIPS.cc/2023/Conference — NeurIPS 2023 poster_

### Official Review · Reviewer_tpKF · 2023-07-02

**Soundness:** 3 good
**Presentation:** 3 good
**Contribution:** 3 good
**Rating:** 6
**Confidence:** 3

**Summary:**

The paper presents a coding scheme for greedy rejection sampling by Harsha, which builds upon the existing A* coding framework. The trick here is to add a partition operation between each reject/accept steps, which allows a more efficient scheme under certain assumptions. Overall, the work contributes meaningfully to the field of rejection sampling.  Minor adjustments to the figure margins are needed for improved readability.


**Strengths:**

The proposed method and analysis demonstrate the potential for achieving faster runtime under certain assumptions about the proposal and target distributions.  The experiment shows the benefits of the method.
The paper is relatively easy to follow.


**Weaknesses:**

While the paper is generally well-written, the lack of sketch proof hinders the ease of understanding why the proposed coding scheme works or how the assumption matters. Including a sketch proof would greatly enhance the understanding of the presented approach.

The differences between the present algorithm and prior work on A* coding must be explained more clearly.

It would be beneficial if the author could comment on the complexity of integrating equation (2) with distributions beyond the Gaussian, thus expanding the scope of the analysis.


**Questions:**

Can the authors explain the implication of assumption 2&3? Comparing the A* coding, I do not follow how GRCS and GRCD can outperform A* coding in terms of runtime complexity.

 Line 274: I do not see where in the appendix factor 2 (in the log term) does not appear in the coding length. Also, 11 bits in O(1) seem to be longer than the bound by Li and El Gamal but A* coding and GRC seem to be on par in Figure 4 (right). Can the author provide some additional justifications for this?

---

> ### Author Rebuttal · Authors · 2023-08-09
>
> We thank TPKF for their time reviewing our paper and their feedback. We were pleased to hear TPFK thought our work was a meaningful contribution and our paper well-written and relatively easy to follow. TPKF raised a few points which we would like to address below.
>
> __Sketch proof:__ We agree the paper would benefit from sketch proofs. We would also like to bring Appendix B to the author’s attention, where we provide an outline of the correctness proof (Theorem 1). We will edit that section to explain how we use assumptions 1, 2, and 3. Here, we provide some intuition as to why GRCS improves over the runtime of GRCG and A* coding and will include this with a sketch proof in our camera-ready version (see also our response to reviewer T3AN).
>
> In GRCG (see Figure 2), the termination probability at each step (given that termination has not yet occurred) becomes smaller, resulting in long runtimes. This is because most proposed samples occur in regions with no remaining mass (white area) under $Q$ and are always rejected. The idea behind GRCS is to make the proposal distribution adapt to the remaining mass. We can achieve this, e.g., with the sample-splitting process.
>
> If $dQ / dP$ is unimodal, then when GRCS rejects a sample, all the mass that has not been ruled out (white area) lies either to the left or right of the rejected sample. Therefore, we can split the sample space into two parts and draw our next sample from the restriction of $P$ to the part containing the remaining white mass. This adapts the proposal’s shape, increasing the termination probability at each step and dramatically improving the algorithm’s runtime. In Appendix C, we formalize this argument and show that the adaptive proposal of GRCS shrinks, homing into the remaining mass exponentially quickly, resulting in far shorter runtimes than GRCG and A* coding (see also the “differences to A* coding” section below).
>
> __Differences to A* coding:__ We thank TPKF for pointing out that the differences between GRC and A* coding could be explained further. Figure 1 illustrates the relationships between the algorithms pictorially. Following TPKF’s advice, we will add a supplementary section explaining these differences in more detail in our camera-ready version and highlight some below.
>
> While both algorithms use partitioning schemes to speed up termination, A* coding is based on Gumbel processes and uses a branch-and-bound search and termination condition. On the other hand, GRC is based entirely on rejection sampling. In fact, it is partly due to the different termination conditions that GRC achieves faster runtimes: GRC terminates once it accepts a sample (i.e., once it produces a good candidate sample), while A* coding must keep going even after it has produced a good candidate until it has sufficiently reduced an upper bound used in its termination condition. This upper bound depends on $D_{\\infty}[Q || P]$, which can be arbitrarily large for a fixed $D_{KL}[Q || P]$. For this reason, A* coding can take much longer to terminate compared to GRC, which can be much faster.
>
> __Integrating equation 2:__ Equation 2 can be integrated efficiently so long as we have access to the CDF of $P$ and $Q$. This CDF is tractable for a wide range of distributions used in learnt compression, including Gaussians, Laplace, uniforms, and many others. While we highlight the access to a CDF as an important requirement in our paper, this is not a significant limitation since most state-of-the-art learnt compression pipelines use tractable distributions such as the above. In our submitted code, we have included Laplace and uniform GRC samplers and executable examples under `notebooks/sampling.ipynb`.
>
> __Q: assumptions 2 & 3:__ Note that assumptions 2 & 3 are only used in Theorem 1 to prove the correctness of the algorithms. They are not used in our proof for the runtime of GRCS (Theorem 3). GRCS achieves faster runtimes than A* coding due to the algorithms’ different constructions and termination conditions. We have outlined how this results in faster runtimes for GRC in the “sketch proof” and “differences to A* coding” sections above.
>
> __Q: codelength bounds:__ The reason the factor of 2 does not appear in eq. 274 is as follows. In the average case, we can use $\\zeta$-coding (which is not possible in the single-shot REC setting) to reduce the prefix code’s length further. Li and El Gamal showed that this could improve the prefix codelength, eliminating the factor of 2 from the bound. We appreciate that this point might not have been clear and will add a clarifying comment to our camera-ready version.
>
> Regarding the constant overhead, note that the quantity involving the 11 bits is an _upper bound_ rather than an equality for the exact codelength. This constant term is likely not incurred in practice: it results from our analysis and can likely be improved. However, the primary purpose of this bound is to show the codelength is proportional to $D_{KL}[Q || P]$. Please also note that the results shown in Figure 4 (right) “do not include additional logarithmic overhead terms,” as explained in the figure caption (see eq. 194). We chose this presentation to have comparable results with A* coding (Flamich et al.). We hope this clarifies the above point.
>
> __Conclusion:__ We thank T3PN for their time and feedback. T3PN generally viewed our work favourably and suggested useful ways to improve our paper. We have highlighted our planned changes to our paper toward this end. If our response and suggested changes satisfy T3PN, we would like to invite them to consider increasing our score.

---

> > ### Comment · Reviewer_tpKF · 2023-08-21
> >
> > Thank you for the response and clarifying the questions. I will keep my score to 6.

---

### Official Review · Reviewer_BHnG · 2023-07-07

**Soundness:** 3 good
**Presentation:** 3 good
**Contribution:** 3 good
**Rating:** 5
**Confidence:** 1

**Summary:**

The authors propose a greedy rejection coding based algorithm for performing compression. The authors claim that GRC an ERC based compression method is the first method which does not make unrealistic assumptions, is guaranteed to finish in fixed amount of time. They generalize the algorithm to arbitrary partition spaces and provides guarantees to finish in limited runtime.

Disclaimer - The reviewer is not the best person to evaluate this paper. They have no experience in this area.



**Strengths:**

* The paper is reasonably well write and is able to explain the challenges very well.

* The proofs by and large look correct, however the reviewer can not comment on their novelty

*The authors have some experiments.


**Weaknesses:**

It would have been good to see how their method works at even higher dimension., can they compress say other models using their method.


**Questions:**

Look at weakness sections

**Limitations:**

The authors have highlighted some limitations.

---

> ### Author Rebuttal · Authors · 2023-08-09
>
> We would like to thank reviewer BHNG for their time to review our work and their feedback. We were happy to hear that the reviewer thought our paper was well-written, discussed relevant challenges well, and was satisfied by the correctness of our proofs. The only point of weakness they mentioned was that they would have liked to see how our approach would perform in even higher dimensions and different models. We would like to respond to this point below.
>
> __Performance in higher dimensions:__ On one hand, it would certainly be interesting and valuable to assess the performance of GRC with larger latent variable models within realistic state-of-the-art compression pipelines. On the other hand, in our view, the main contribution of our paper is a theoretical one and consists of (1) devising GRC, (2) proving that it terminates and yields unbiased samples, and (3) achieving optimal compression codelengths and near-optimal runtimes under realistic assumptions. Our experiments with VAE-based compression constitute a proof-of-concept experiment to validate the efficacy of GRC in a controlled setting rather than to integrate it within a realistic compression pipeline with high-dimensional latent variable models.
>
> Developing a realistic compression pipeline with large latent variable models using GRC is an exciting direction of empirical work. However, it is beyond the scope of our current investigation, which is chiefly theoretical. For this reason, we think our work should not be viewed negatively on this basis.
>
> __Conclusion:__ We thank reviewer BHNG for the time taken to review our paper. We hope we have adequately addressed their concern regarding compression with higher-dimensional distributions. We note that while reviewer BHNG found no substantial issues with our paper and was overall satisfied by the quality of our exposition and the correctness of our results, they have recommended our paper with a somewhat conservative acceptance score. In light of this, we would like to invite reviewer BHNG to consider raising their score if they are satisfied with and convinced by our response to their review, as well as the other reviews and subsequent discussions.

---

### Official Review · Reviewer_ULav · 2023-07-17

**Soundness:** 3 good
**Presentation:** 3 good
**Contribution:** 3 good
**Rating:** 5
**Confidence:** 1

**Summary:**

This paper presents a new algorithm, Greedy Rejection Coding (GRC), for Relative Entropy Coding (REC). REC is an alternative to entropy coding and quantization, which naturally interfaces with continuous variables and has applications in data compression. Current REC algorithms have faced challenges like slow runtimes or restrictive assumptions, hindering their widespread use. The new GRC method, along with its two variants GRCS and GRCD, generalizes a previous rejection sampling-based algorithm to arbitrary probability spaces and partitioning schemes. The authors demonstrate that GRC terminates almost surely, producing unbiased samples, and that GRCS has an optimal expected runtime and codelength for a wide range of one-dimensional problems. The effectiveness of GRC is showcased in a compression pipeline with variational autoencoders on the MNIST dataset, where it outperforms in compression efficiency.

**Strengths:**

1. The paper introduces a new approach, Greedy Rejection Coding (GRC), to the field of rate-distortion compression (REC). The authors build on the rejection algorithm from Harsha et al., extending it to arbitrary probability spaces and partitioning processes. Theoretical results show that the GRC method is accurate and terminates successfully under certain general assumptions.

2. The authors validate their ideas with a proof-of-concept implementation using Variational Autoencoders (VAEs) on the MNIST dataset. They compare them against previous methods like A* coding. Their experiments validate the theoretical underpinnings of the algorithms and show improvements over prior work.

3. The paper deals with the challenge of coding overhead in compression problems. The authors propose a principled modification to the Evidence Lower Bound (ELBO) and entropy coding of GRCD's indices using a ζ distribution as innovative ways to improve compression efficiency.

**Weaknesses:**

1. **Dependency on CDF:** A major limitation of the proposed Greedy Rejection Coding (GRC) approach is that it requires the ability to evaluate the cumulative distribution function (CDF) of the distribution Q. This could potentially limit the algorithm's practicality in situations where the CDF is difficult or computationally expensive to calculate.

2. **Applicability to Multivariate Distributions:** The paper mentions that the current approaches assume target-proposal pairs over the real line. This might limit the effectiveness of the proposed algorithm when dealing with multivariate distributions, as they are decomposed into univariate conditionals, resulting in additional coding overhead per dimension.

3. **Lack of Generalizability to Other Datasets:** The empirical validation of the approach is based on synthetic data and one real-world dataset (MNIST). How well the proposed approach would generalize to more complex datasets or tasks is unclear.

4. **Overhead Mitigation Measures:** The proposed methods to reduce the overhead, such as a modified ELBO and entropy coding using a ζ distribution, could add complexity to the algorithm. Moreover, it's not yet fully clear how much improvement they can bring in diverse scenarios.

5. **Efficiency Analysis:** While the paper discusses the runtime of the proposed algorithm compared to existing methods, it does not provide a detailed computational efficiency analysis. Therefore, it might be unclear how the proposed algorithm would perform under different computational resources.

**Questions:**

Please see the Weaknesses.

**Limitations:**

This work does not pose any ethical concerns.

---

> ### Author Rebuttal · Authors · 2023-08-09
>
> We thank reviewer ULAV for their time reviewing our paper. We are happy they found our theoretical and experimental results as strong points. Reviewer ULAV also raised some concerns, which we address below.
>
> __Dependency on CDF:__ ULAV commented that since GRC requires access to the CDF of the target $Q$ and proposal $P$, this may “limit the algorithm's practicality in situations where the CDF is difficult or computationally expensive to calculate.”
>
> In many state-of-the-art communication pipelines, $P$ and $Q$ have simple forms, and their CDFs can be evaluated cheaply:
>
> 1. In data compression, e.g., using VAEs, $P$ and $Q$ are typically chosen to be uniform, Gaussian, or Laplace.
> 2. In model compression, both $P$ and $Q$ are often Gaussian.
> 3. In federated learning, $P$ and $Q$ are also typically Gaussian.
>
> The CDFs of most base distributions in learnt compression are easy to evaluate. Existing pipelines __use tractable distributions to evaluate their training objectives__, and we expect future ones will do so too. Therefore, assuming access to the CDF is not a constraining assumption.
>
> __Applicability to multivariate distributions:__ ULAV commented that GRC assumes $P$ and $Q$ over the reals, which might limit its effectiveness when decomposing joints into univariate conditionals due to the additional coding overheads. We do not entirely agree with this comment:
>
> 1. GRC directly applies to multivariate $P$ and $Q$. However, we have yet to find appropriate assumptions and a partitioning process that simultaneously achieves optimal runtimes and codelengths in this setting. While this problem is challenging, we believe more suitable partitioning schemes may exist, so we do not view this as a fundamental limitation of GRC.
> 2. The provably fast instances of A* coding, the current state-of-the-art, also assume univariate $P$ and $Q$. Therefore, this does not limit the applicability of GRC any more than the current state-of-the-art.
> 3. We have provided practical approaches to mitigate overhead codelengths, showing these can be as low as 6% of the total coding cost. We regard these approaches as proofs-of-concept and believe that future work could further reduce these overheads.
>
> Overall, we believe this is neither a prohibitive nor a fundamental drawback of GRC, and we do not think our work should be viewed unfavourably on these grounds.
>
> __Lack of generalisability to other datasets:__ Reviewer ULAV commented that it is unclear whether our approach would generalize to more complex datasets or tasks.
>
> We stress that our main contributions are theoretical rather than empirical and note that applying REC to complex learned compression pipelines is well outside the scope of this work. Moreover, we do not expect the performance of REC would differ on larger-scale problems: Flamich et al. (2020) studied applying REC for high-resolution image compression using VAEs and found that their method was competitive with the state-of-the-art at the time.
>
> Nevertheless, our runtime and codelength bounds apply directly to a range of existing and future pipelines. These give quantitative guarantees about the performance of GRC as a function of the target and proposal, which are aspects of the model alone. This is one of the appeals of GRC (and generally REC): given a model’s performance on some data, one already has a good idea of the performance of the entire pipeline.
>
> __Overhead mitigation measures:__ One issue raised by ULAV is that our proposed methods for mitigating overhead costs could add complexity to the algorithm and that it is “not yet fully clear how much improvement these can bring in diverse scenarios.” We would like to respond with the following points:
>
> 1. Our modified ELBO objective is a simple, efficient, and principled method: it modifies the objective to account for the overhead codes during training, reducing overheads and incurring negligible computational overheads in practice.
> 2. $\\zeta$-coding is a widely known method already applied in REC (see Li and El Gamal, 2017). Using $\\zeta$-coding is a straightforward, computationally cheap procedure that does not significantly complicate the overall compression pipeline, especially considering the improvements it yields.
> 3. As demonstrated in our experiments, as the number of dimensions in $Q$ and $P$ increases, the performance gains from using the methods above also increase. This suggests our methods would be highly beneficial for larger latent variable models across a range of scenarios.
>
> We believe the aforementioned methods are simple, principled, computationally cheap, effective in practice, and arguably of interest to related work. Further, they bear minimal complexity, which is far outweighed by their improvements.
>
> Efficiency analysis: ULAV commented that our paper “does not provide a detailed computational efficiency analysis.” We think this comment somewhat misrepresents our work. There are two measures of computational efficiency: computational and memory complexity. Runtime is synonymous with computational complexity, and we have provided near-optimal bounds for the expected runtime of GRCS. Further, at each step, GRC updates all variables in-place (Algorithm 2), so its memory cost is O(1). We will clarify this in our camera-ready version.
>
> Could the reviewer elaborate on what additional computational analysis they would like to see? If there are no further points of analysis they can think of, we kindly ask them to reconsider this point.
>
> __Conclusion:__ We thank ULAV for their time and feedback. Overall we found that some comments did not accurately represent our work. We hope our response has clarified these points and sufficiently addressed their concerns. If ULAV thinks we have adequately addressed their concerns, we would like to invite them to consider raising their score.

---

### Official Review · Reviewer_t3AN · 2023-07-26

**Soundness:** 4 excellent
**Presentation:** 4 excellent
**Contribution:** 3 good
**Rating:** 6
**Confidence:** 3

**Summary:**

This paper proposes new Greedy Rejection Coding methods that generalize to continuous probability space and arbitrary partitioning schemes. The new methods produce correct samples with the expected code length of $O(D_{KL}[Q||P])$ where $Q$ is the target distribution and $P$ is the proposal distribution. The paper then evaluates the new algorithms and verifies the improvements in runtime in practical experiments.

**Strengths:**

- The proposed methods achieve the optimal runtime up to a constant order while guaranteeing the optimal expected codelength.

- The experiment results also verify the theory. The runtime of proposed methods (in terms of the number of steps) stays constant when $D_\infty[Q||P]$ varies and is always smaller than the runtimes of previous methods AS* and AD*.

- Comparisons to previous results, technical novelties, and limitations are thoroughly discussed.

- The paper also provides plenty of high-level discussions and visualizations of their results.



**Weaknesses:**

- I think some intuitions about why the modifications help improve the runtime and also why the authors decide to make that particular modification can be added to the main text to improve the readability of the paper.

**Questions:**

- Can the author explain why the bound mentioned in line 274 is better than the bound in eq. 10?

- What is the wall-clock runtime of the proposed methods? It's clear that the proposed methods significantly improve the runtime in the number of steps but I'm curious if some of the extra steps in the algorithms could cause overhead in terms of wall-clock runtime/step.

- There's a typo in line 321: "GRC can become significant when the KL over Hence..."

---

> ### Author Rebuttal · Authors · 2023-08-09
>
> We thank reviewer T3AN for their time reviewing our paper and valuable feedback. We were happy to hear they found our theoretical and experimental results as strong points of this work. We were also pleased that they thought our exposition, discussion of related work and limitations, and vizualisations were of a good standard. T3AN also raised a point on the intuition behind our suggested modifications and a few questions we address below.
>
> __Intuition on the effectiveness of our modifications:__ T3AN commented that we could improve the paper's readability by adding some intuitions about why our proposed modifications help speed up the runtime.
>
> We agree with this point and will amend our paper based on the argument we outline below:
>
> First, let us explain the motivation for making the particular modifications that we chose in this paper. We already know from a result by Augustsson & Theis (discussed in our paper) that we cannot avoid exponential worst-case runtimes in REC without further assumptions in $P$ and $Q$. Therefore, we need additional assumptions to achieve a practically fast algorithm. Flamich et al. 2022 showed that assuming $Q$ and $P$ over the reals and a unimodal density ratio $dQ / dP$, and using a sample splitting scheme results in a dramatic speedup and a computationally tractable algorithm compared to a global partitioning scheme (i.e. compared to no partitioning). The success of this approach motivated us to examine if we can extend similar partitioning schemes to rejection-based coders and quantify the speedup we can obtain under the same assumptions.
>
> We now explain why these modifications improve the runtime of the algorithm. In regular GRCG (figure 2), the probability of termination (= (green area) / (green area + white area)) becomes smaller as the algorithm progresses. Intuitively, this happens because most of the proposal $P$ proposes samples in regions with no remaining mass under $Q$. These samples will never get accepted, causing long runtimes. If we can make the algorithm somehow adaptive so that $P$ adapts to the shape of the white area, we can reduce wasteful samples and speed up termination. We can achieve this using the sample-splitting process.
>
> Note that if $dQ / dP$ is unimodal, then when GRC rejects a sample, all the mass that has not been ruled out (white area) lies either to the left or to the right of the rejected sample (see Figure 2; left subfigure: all the mass lies to the right of sample #1; middle subfigure: all the mass lies to the left of sample #2). Therefore, we can split the sample space into two parts and draw our next sample from the restriction of $P$ to the part containing all the remaining white mass. This increases the termination probability at each step, remarkably improving the algorithm’s runtime to be order-optimal. This notion extends to other adaptive partitioning schemes, such as the dyadic partitioning process. In this case, we have to choose the partition in a randomized way (see Algorithm 2) to ensure the samples are unbiased.
>
> __We will add this further intuition in our camera-ready version.__
>
> __Improved bound:__ The reason why the bound in line 274 improves on eq 10 is as follows. First, in the average case, we assume a publicly available joint distribution $P_{X, Y}$ over the correlated variables $X$ and $Y$. Given a private sample $Y \\sim P_Y$, the encoder aims to communicate a sample from the conditional $X \\sim P_{X | Y}$. Therefore in this case, if for a given $Y$, we set $P = P_X$ and $Q = P_{X | Y}$ and use a GRC, we can communicate a sample $X \\sim P_{X | Y}$ using at most
>
> $$D_{KL}[Q || P] + 2 \\log(D_{KL}[Q || P] + 1) + \\mathcal{O}(1) = D_{KL}[P_{X | Y} || P_X] + 2 \\log(D_{KL}[P_{X | Y} || P_X] + 1) + \\mathcal{O}(1)$$
>
> bits on average. Further, averaging out over $Y \\sim P_Y$, we obtain an average-case communication cost of at most
>
> $$\\mathbb{I}[X; Y] + 2 \\log(\\mathbb{I}[X; Y] + 1) + \\mathcal{O}(1)$$
>
> bits. Lastly, we can use $\\zeta$-coding (which is not possible in the single-shot REC setting) to further improve the coefficient in front of the logarithm, to improve the communication cost to at most
>
> $$\\mathbb{I}[X; Y] + \\log(\\mathbb{I}[X; Y] + 1) + \\mathcal{O}(1).$$
>
> bits. We hope this clarifies the question raised by T3AN and welcome further clarifying questions. We will add a discussion of this point in our camera-ready version.
>
> __Runtimes:__ Although the modified algorithm involves a few additional steps, these are far outweighed by the reduction in the overall number of steps they yield. Specifically using our JAX implementation, choosing $Q = \\mathcal{N}(0.5, 0.8^2)$ and $P = \\mathcal{N}(0, 1)$ and drawing $10^4$ samples with each method, yields wallclock times of $3.24$ sec, $0.90$ sec and $0.49$ for GRCG, GRCS and GRCD respectively (where for GRCG we have used algorithm 1 instead of algorithm 2 to avoid unnecessary computations). The difference in runtimes can be made arbitrarily larger by picking problems with a fixed KL but increasing infinity divergence. Note that although the modified algorithms require slightly more computation per step, they are much faster because they take far fewer steps. The exact runtimes above are implementation-dependent though we believe our implementation is relatively well-optimized and fair between the methods.
>
> __Typo:__ Thank you for pointing this out. We have updated this in our working draft.
>
> __Conclusion:__ We thank T3AN for their time and valuable feedback. We found that their review was positive, and we hope our rebuttal has addressed the points they raised. If T3AN is satisfied with our response, we would like to invite them to consider increasing our score.

---

> > ### Comment · Reviewer_t3AN · 2023-08-19
> >
> > Thanks for the response! I will keep my score.

---

### Author Rebuttal · Authors · 2023-08-09

Dear Reviewers and ACs,

We would like to thank all four reviewers for the time they took reviewing our paper and for their valuable feedback. We found that the feedback from all four reviewers was generally positive. Most reviewers listed our theoretical contributions and the experimental validation of our approach among the main strengths of our work. They also thought that our paper was generally well-written and our exposition to be of a good standard.

Some of the reviewers also raised a few points of concern and some questions including, for example, the tractability of the CDFs used within GRC and technical clarifications on our derived bounds. In our view, we thoroughly addressed their points of concern in our response, justifying why these are not significant limitations of our method and clarifying their questions. The reviewers also gave useful suggestions to further improve our paper, such as adding more sketch proofs and discussions to provide additional intuition about our algorithms. In response, we provided an overview of concrete changes we plan to make in our manuscript for this purpose. We think these changes are beneficial and thank the reviewers for suggesting them.

In conclusion, we think the reviews for our work were overall positive. We hope that we have addressed the reviewers’ points of concern and their questions to their satisfaction and welcome further feedback and discussion on our work.



Thank you,

the authors of paper 5337

---

### Author Response · Authors · 2023-08-17
**Request for the Reviewers to Respond to Our Rebuttal**

We thank the reviewers again for the feedback they provided on our work. As the author-reviewer discussion period ends on 21 August, we kindly ask the reviewers to let us know if our rebuttal has addressed their concerns adequately as soon as possible so we may respond to any outstanding issues on time. Additionally, if our rebuttal has effectively addressed the reviewers’ concerns, we kindly invite them to acknowledge it and consider raising their scores. Thank you once again for your time and feedback.

Thank you,

Authors of paper 5337

---

### Decision · Program_Chairs · 2023-09-21

**Decision:**

Accept (poster)

**Comment:**

This paper presents a new algorithm for efficient compression of real-valued data, along with theoretical analysis and preliminary empirical validation. The authors are suggested to work on the presentation of their methods: the reviewers seemed overall to find the paper somewhat difficult to parse. However overall the paper was deemed to present a valuable contribution to an important problem.